# A Unified Objective for On-Policy Reinforcement Learning in Stationary and Non-Stationary Environments

## Abstract

A fundamental dichotomy between the discounted and average return has long existed in the field of deep reinforcement learning (DRL). Algorithms based on the average return assume the existence of stationary state distribution and often struggle in non-stationary or episodic settings. In contrast, algorithms optimizing the discounted return are well-suited for non-stationary tasks but may learn suboptimal policies in long-term stationary settings due to the inherent bias introduced by the discount factor. This forces practitioners to select an objective based on the specific environment, thereby limiting the development of general and robust DRL algorithms. We introduce the $k$-**sliding-window return**, a novel objective that bridges these two criteria. We instantiate this concept with a practical on-policy algorithm, $k$-sliding-window PPO ($k$SW-PPO). Besides, we provide theoretical analysis showing that the loss of our objective converges to that of the average return while maintaining a bounded bias relative to the discounted return. We validate our claims through experiments on a suite of MuJoCo continuous control tasks. The results demonstrate that $k$SW-PPO achieves performance competitive with average-return PPO in stationary environments, while matching the performance of its discounted-return counterpart in non-stationary settings. Our results establish the $k$-sliding-window return as a unified objective that eliminates the need for an a priori choice between discounting and averaging, which we hope to inspire the development of more robust and general-purpose DRL algorithms. Code is available at `https://anonymous.4open.science/r/kSW-PPO-1E7C`.

## 1 Introduction

Deep Reinforcement Learning (DRL) has achieved remarkable success in a wide range of applications, such as gaming (Perolat et al. (2022); Qi et al. (2023); Li et al. (2025)), combinatorial optimization (Berto et al. (2025); Kallestad et al. (2023)), recommendation systems (Zhao et al. (2021); Afsar et al. (2022)), and robotics (Franceschetti et al. (2021); Raffin et al. (2022)). In these applications, the environment is typically modeled as a Markov Decision Process (MDP), where the goal is to find an optimal policy that maximizes the total return. However, to ensure training stability, surrogate objectives are commonly employed instead of directly optimizing the total return. The most prevalent objectives are the discounted return Bellman (1966) and the average return Blackwell (1962).

However, these two objectives each possess distinct strengths and weaknesses, which leads to a fundamental performance dichotomy. On one hand, the discounted return is widely applicable because its well-definedness does not rely on any special assumptions about the environment. Yet, the introduction of the discount factor $\gamma$ inherently biases the objective towards short-term performance, which can result in suboptimal policies for long-term stationary tasks (Zhang & Ross (2021)). On the other hand, the average return is specifically designed for long-term settings. A critical limitation, however, is its theoretical reliance on the existence of a stationary state distribution induced by a policy. This assumption is often violated in practical scenarios, such as episodic tasks or environments without a strong mixing property, which leads to poor algorithmic performance.

We empirically illustrate this dichotomy and trace its root cause to the underlying environment structure. Figures 1(a) analyzes the state visitation patterns of a near-optimal policy in two MuJoCo

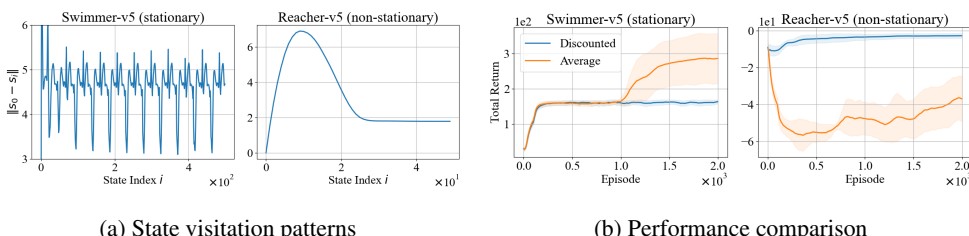

(a) State visitation patterns            (b) Performance comparison

Figure 1: Performance dichotomy between average and discounted return. Each plot in (a) shows the Euclidean distance from the initial state, $\|s_i - s_0\|$, as a function of the time step $i$ within a trajectory sampled from a near-optimal policy. Plots in (b) confirm the performance dichotomy.

environments: *Swimmer* and *Reacher*. The clear **periodicity** in *Swimmer* indicates a stationary state distribution, while the **aperiodic**, divergent pattern in *Reacher* signifies a non-stationary one. As a direct consequence, Figure 1(b) confirms that the performance of the two objectives diverges based on this structure: average-return PPO excels in the stationary *Swimmer* environment, whereas discounted-return PPO is superior in the non-stationary *Reacher*.

Consequently, practitioners face a difficult choice: the best objective depends on the environment, which is often unknown a priori. This requirement for environment-specific tuning limits the robustness and generality of DRL algorithms. This motivates our central research question:

*Is it possible to design a single unified objective that consistently delivers strong performance in both stationary and non-stationary environments?*

To address this question, we propose the $k$-sliding-window return, a unified objective that synthesizes the advantages of both objectives. It is defined as the undiscounted sum of rewards within a $k$-step window that slides along a potentially infinite-horizon trajectory. This design offers a distinct advantage: unlike the discounted return, its lack of discounting prevents the introduction of bias in stationary tasks; unlike the average return, it does not rely on the existence of a stationary distribution. We theoretically validate these benefits. First, we establish an upper bound on the loss difference relative to the discounted return. We show this bound is minimized when $k$ is set to $1/(1-\gamma)$. Second, we prove that the loss difference relative to the average return shrinks as the policy converges. These theoretical strengths translate into practice: experiments on MuJoCo show that our single objective is robust and performs competitively across both stationary and non-stationary environments, which demonstrates its superiority as a general-purpose objective.

Our work makes the following key contributions:

(1) We introduce the $k$**-sliding-window return**, a novel and unified objective that bridges the gap between discounted and average return in on-policy RL. Based on this, we develop $k$SW-PPO, a practical algorithm that is simple to implement yet effectively adaptable to diverse environments.

(2) We establish a firm theoretical foundation, which proves that the loss derived from our objective converges to that of the average return as the policy improves, while simultaneously maintaining a bounded difference from the discounted return. This provides the theoretical justification for a single objective that performs robustly in both stationary and non-stationary regimes.

(3) We demonstrate empirically that $k$SW-PPO effectively unifies performance across environments. It seamlessly combines the strengths of both approaches: it achieves state-of-the-art performance comparable to average-return PPO in stationary environments, while simultaneously maintaining robust performance rivaling discounted-return PPO in non-stationary environments.

## 2    RELATED WORKS

The optimization objective is a cornerstone of RL, fundamentally shaping an agent's behavior. In infinite-horizon settings, the total return is an ideal but often intractable objective. Consequently, the field has largely converged on two surrogate objectives: the discounted return and the average

return. The relationship between these objectives, their theoretical properties, and their empirical performance have been subjects of enduring research interest.

The discounted return remains the most widely adopted objective, prized for its mathematical tractability and general applicability. A substantial body of work is dedicated to understanding the empirical impact of the discount factor, $\gamma$. It has been shown to act as a form of $L_2$ regularization on the value function (Amit et al. (2020)), and smaller values of $\gamma$ can lead to better policies on smaller datasets in offline RL (Hu et al. (2022)). A second thread of research seeks to understand and overcome the training instability that arises as $\gamma \to 1$. The fragmentation of the policy's loss landscape (Wang et al. (2023)) and the existence of state cycles in the environment (Gao et al. (2022)) have been identified as significant sources of this optimization difficulty. Several approaches have been proposed to mitigate this instability. For instance, Grand-Clément & Petrik (2023) leverage the Blackwell optimality criterion to theoretically derive an optimal range for $\gamma$. Kim et al. (2022) propose a novel method that adjusts $\gamma$ based on environmental uncertainty to improve robustness. Other techniques designed for on-policy algorithm include using Taylor expansions to approximate the value function for a target $\gamma$ from a smaller one (Tang et al. (2021)) and redefining the value function to excise problematic state cycles, thereby increasing training stability (Gao et al. (2022)).

The average return was developed as an alternative to eliminate the reliance on $\gamma$ and its associated bias. Research in this area spans both theoretical and practical developments. On the theoretical side, a key focus has been to formalize the connection between the average and discounted returns. Kakade (2001) established performance bounds for policies optimized under the discounted criterion but evaluated under the average criterion. Subsequent analyses demonstrated the asymptotic convergence between the discounted and average value functions under specific conditions, such as for linear models (Tsitsiklis & Van Roy (2002)) or in the limit of the horizon length (Hutter (2006)). More recently, Siddique et al. (2020) extended the work of Kakade (2001) to the non-linear function approximation setting. On the practical side, replacing the discounted return with the average return has inspired numerous variants of popular DRL algorithms, such as APO (Ma et al. (2021), based on PPO (Schulman et al. (2017))), ATRPO (Zhang & Ross (2021), based on TRPO (Schulman et al. (2015))), ARO-DDPG (Saxena et al. (2023), based on DDPG (Lillicrap et al. (2016))), and RVI-SAC (Hisaki & Ono (2024), based on SAC (Haarnoja et al. (2018))). These algorithms have consistently shown superior performance compared to their discounted-return counterparts in long-term stationary tasks.

While the existing literature provides a comprehensive analysis of the algorithmic and theoretical trade-offs between these two objectives, a systematic investigation into how the inherent properties of the environment might dictate this trade-off is notably absent. Our work seeks to address this gap.

## 3 PRELIMINARIES

A MDP is defined by the tuple $(S, A, P, r, \gamma)$. $S$ is the state set, with each $s \in S$ representing a state. $A$ is the action set, with each $a \in A$ representing an action. $P : S \times A \to \mathcal{P}(s)$ is the Markov transition kernel and $P(\cdot|s, a)$ is the probability for next state. $r : S \times A \to [R_{\min}, R_{\max}]$ is the reward function. A policy $\pi : S \to \mathcal{P}(A)$ maps any $s \in S$ to a probability distribution $\pi(\cdot|s)$ over $A$. Without loss of generality, this work regards $S$ and $A$ as countable sets.

The transition kernel under a policy $\pi$ is denoted as $P^\pi : S \to \mathcal{P}(S)$, where $\mathcal{P}(S)$ is the probability defined on $S$ and $P^\pi(s'|s) = \sum_{a \in A} \pi(a|s)P(s'|s, a)\,da$. The $t$-step transition probability from $s$ to $s'$ is defined as $P_t^\pi(s'|s) = \sum_{s''} P_{t-1}(s''|s)P(s'|s'')$ and $P_1^\pi(s'|s) = P^\pi(s'|s)$.

Throughout our analysis, the distance between two distributions $\mu$ and $\nu$ over $S$ is measured by the total variation (TV) distance, $D_{\mathrm{TV}}(\mu(\cdot), \nu(\cdot)) := \frac{1}{2} \sum_{s \in S} |\mu(s) - \nu(s)|$.

### 3.1 DISCOUNTED RETURN

Given discount factor $\gamma \in (0, 1)$, the corresponding discounted state value $V_\gamma^\pi(\cdot)$, the discounted state-action value $Q_\gamma^\pi(\cdot, \cdot)$ and the discounted advantage $A_\gamma^\pi(\cdot, \cdot)$, are defined as:

$$V_\gamma^\pi(s) = \mathbb{E}_{\pi, P}\left[\sum_{t=0}^\infty \gamma^t r(s_t, a_t)\bigg| s_0 = s\right], \qquad Q_\gamma^\pi(s, a) = \mathbb{E}_{\pi, P}\left[\sum_{t=0}^\infty \gamma^t r(s_t, a_t)\bigg| s_0 = s, a_0 = a\right],$$

$$\text{and } A_\gamma^\pi(s,a) = Q_\gamma^\pi(s,a) - V_\gamma^\pi(s) = r(s,a) + \mathbb{E}_{s'\sim P(\cdot|s,a)}\left[V_\gamma^\pi(s')\right] - V_\gamma^\pi(s).$$

The probability distribution $d_s^\pi(\cdot)$ is defined as $d_s^\pi(s') = (1-\gamma)\sum_{k=0}^\infty \gamma^t P_k^\pi(s'|s)$.

## 3.2 AVERAGE RETURN

**Assumption 1** (Ergodic(Meyn & Tweedie (2012))). *Suppose for every stationary policy $\pi$, the induced Markov chain with transition probability $P^\pi$ is reducible and aperiodic.*

Assumption 1 ensures that the Markov chain induced by any policy $\pi$ is ergodic. This guarantees the existence and uniqueness of a stationary distribution $d^\pi(\cdot)$ over the state space (Meyn & Tweedie (2012)). This ergodicity is a standard condition for the average return to be well-defined. The gain (or average reward) for policy $\pi$ is then defined as the expected reward under this stationary distribution, which is $\rho^\pi = \mathbb{E}_{s\sim d^\pi, a\sim \pi(\cdot|s)}\left[r(s,a)\right]$. Then all the value functions for a policy $\pi$ are defined as:

$$\bar{V}^\pi(s) = \mathbb{E}_{\pi,P}\left[\sum_{t=0}^\infty (r(s_t,a_t) - \rho^\pi)\bigg|s_0 = s\right],$$

$$\bar{Q}^\pi(s,a) = \mathbb{E}_{\pi,P}\left[\sum_{t=0}^\infty (r(s_t,a_t) - \rho^\pi)\bigg|s_0 = s, a_0 = a\right],$$

$$\text{and } \bar{A}^\pi(s,a) = \bar{Q}^\pi(s,a) - \bar{V}^\pi(s) = r(s,a) - \rho^\pi + \mathbb{E}_{s'\sim P(\cdot|s,a)}\left[\bar{V}^\pi(s')\right] - \bar{V}^\pi(s).$$

# 4 $k$-SLIDING-WINDOW RETURN

In this section, we formally introduce the $k$-sliding-window return, a novel objective designed to deliver strong performance in both stationary and non-stationary environments for on-policy DRL. The core idea is to define the objective as the undiscounted sum of rewards within a $k$-step window that slides along a potentially infinite-horizon trajectory.

Specifically, for any state $s$ encountered along the trajectory, its state value function under policy $\pi$ is defined as the expected sum of the next $k$ rewards:

$$V_k^\pi(s) = \mathbb{E}_{P,\pi}\left[\sum_{t=0}^{k-1} r(s_t,a_t)\bigg|s_0 = s\right],$$

The state-action value function is defined as $Q_k^\pi(s,a) = \mathbb{E}_{P,\pi}\left[\sum_{t=0}^{k-1} r(s_t,a_t)|s_0 = s, a_0 = a\right]$ and the advantage function is

$$A_k^\pi(s,a) = Q_k^\pi(s,a) - V_k^\pi(s) = r(s,a) + \mathbb{E}_{s'\sim P(\cdot|s,a)}\left[V_{k-1}^\pi(s')\right] - V_k^\pi(s).$$

A key feature of the $k$-sliding-window return is that while defined over a finite step $k$, it is applied to continuing (infinite-horizon) tasks. This induces a value function that naturally "rolls" with the trajectory. This stands in stark contrast to the total return in episodic MDPs, where value functions are computed backward from a fixed termination point.

## 4.1 $k$-SLIDING-WINDOW PROXIMAL POLICY OPTIMIZATION

Based on the value functions defined above, we introduce our novel on-policy algorithm, namely $k$SW-PPO ($k$-sliding-window PPO). To analyze its theoretical properties, we first introduce a performance difference lemma. This lemma allows us to establish a critical connection between the policy gradient loss of $k$SW-PPO and the resulting difference in policy performance. Central to this analysis is the state visitation distribution over a $k$-step horizon. We formally define this distribution for a policy $\pi$ starting from state $s$ as: $d_{k,s}^\pi(s') = \frac{1}{k}\sum_{t=0}^{k-1} P_t^\pi(s'|s)$.

**Lemma 1** (Performance difference). *For any two policies $\pi$ and $\pi'$, and any initial state $s$, the following holds:*

$$V_k^\pi(s) - V_k^{\pi'}(s) = k \cdot \mathbb{E}_{s'\sim d_{k,s}^\pi, a\sim\pi}\left[A_k^{\pi'}(s',a)\right] + \Delta(s).$$

*where $\Delta(s) = \mathbb{E}_{\pi,P}\left[\sum_{t=1}^{k-1}\mathbb{E}_{\pi',P}[r(s_{t+k-1},a_{t+k-1})|s_t]\big|s_0 = s\right] - \mathbb{E}_{\pi,P}\left[V_{k-1}^{\pi'}(s_k)\big|s_0 = s\right]$.*

The performance difference lemma for $k$-sliding-window return contains an extra term $\Delta(s)$ compared to its counterparts for the discounted (Kakade & Langford (2002)) and average return (Zhang & Ross (2021)), yet $\Delta(s)$ is controlled by the distance between $\pi$ and $\pi'$ for any $s$ (by Lemma 6 in the Appendix). Thus we can still construct the loss for $k$-sliding-window return by advantage function, and the difference between loss optimization and policy improvement is still controlled by the distance between two policies.

**Theorem 1** (Relation between loss and performance difference). *For a policy $\pi_\theta$ and a previous policy $\pi_{\theta_{old}}$, define the loss as:*

$$\mathcal{L}_k(\pi_\theta, \pi_{\theta_{old}}, s) = \mathbb{E}_{s' \sim d^{\pi_{\theta_{old}}}_{k,s}, a \sim \pi_{\theta_{old}}} \left[ \frac{\pi_\theta(a|s')}{\pi_{\theta_{old}}(a|s')} A_k^{\pi_{\theta_{old}}}(s', a) \right].$$

*The relationship between policy improvement and this loss is bounded by:*

$$\left| V_k^{\pi_\theta}(s) - V_k^{\pi_{\theta_{old}}}(s) - k \cdot \mathcal{L}_k(\pi_\theta, \pi_{\theta_{old}}, s) \right| \leq k(k-1) D_{\mathrm{TV}}^{\max}(\pi_{\theta_{old}} \| \pi_\theta)(4\alpha D_{\mathrm{TV}}^{\max}(\pi_{\theta_{old}} \| \pi_\theta) + |R_{\max}|),$$

*where $D_{\mathrm{TV}}^{\max}(\pi_{\theta_{old}} \| \pi_\theta) = \max_s D_{\mathrm{TV}}(\pi_{\theta_{old}}(\cdot|s), \pi_\theta(\cdot|s))$ and $\alpha = \max_{s,a} \left| A_k^{\pi_{\theta_{old}}}(s, a) \right|$.*

By applying Pinsker's inequality (Tsybakov (2008)), the total variation distance in the bound of Theorem 1 can be replaced with the KL divergence, aligning our bound with the standard format of TRPO/PPO. The core implication of Theorem 1 is that for any state $s \in S$, the policy improvement is well-approximated by the loss:

$$V_k^{\pi_\theta}(s) - V_k^{\pi_{\theta_{old}}}(s) \approx k \cdot \mathcal{L}_k(\pi_\theta, \pi_{\theta_{old}}, s),$$

provided that the distance between the new policy $\pi_\theta$ and the old policy $\pi_{\theta_{old}}$ is small. This confirms that maximizing the loss promotes policy improvement. Building upon this loss, we introduce the $k$-sliding-window PPO ($k$SW-PPO) algorithm, which is detailed in Algorithm 1.

---

**Algorithm 1:** $k$SW-PPO

**Input:** initial policy $\pi_0$, initial value function $V_0$, window size $k$, training episode numer $T$
**Output:** learned policy $\pi_T$

1 **for** $t = 1, ..., T$ **do**
2      Sample a trajectory $\{s_0, a_0, ..., s_{n-1}, a_{n-1}, s_n\}$ using $\pi_{t-1}$.
3      Calculate $k$-sliding-window return $G_k^i = \sum_{j=i}^{i+k-1} r(s_j, a_j)$.
4      Calculate advantage $A_{t-1}(s_i, a_i) = G_k^i - V_{t-1}(s_i)$.
5      Policy Improvement: Define $I_{t-1}^i = \frac{\pi(a_i|s_i)}{\pi_{t-1}(a_i|s_i)}$ and learn a new policy through

$$\pi_t = \arg\max_\pi \frac{1}{n} \sum_{i=0}^{n-1} \min \left( I_{t-1}^i A_{t-1}(s_i, a_i), \mathrm{Clip}(I_{t-1}^i, 1-\epsilon, 1+\epsilon) A_{t-1}(s_i, a_i) \right).$$

6      Policy Evaluation: Learn a new value function through

$$V_t = \arg\min_V \frac{1}{n} \sum_{i=0}^{n-1} \left( V(s_i) - G_k^i \right)^2.$$

7 **end**

---

### 4.2 COMPARISON WITH THE LOSS OF DISCOUNTED RETURN

We now quantify the relationship between our objective and the discounted return, showing their corresponding loss functions are maximally similar when $k$ is chosen to be $1/(1-\gamma)$. To analyze the relationship, we formally define the loss for the discounted return (Schulman et al. (2015)) as:

$$\mathcal{L}_\gamma(\pi_\theta, \pi_{\theta_{old}}, s) = \mathbb{E}_{s' \sim d^{\pi_{\theta_{old}}}_s, a \sim \pi_{\theta_{old}}} \left[ \frac{\pi_\theta(a|s')}{\pi_{\theta_{old}}(a|s')} A_\gamma^{\pi_{\theta_{old}}}(s', a) \right],$$

where $d_s^\pi(s') = (1-\gamma) \sum_{t=0}^{\infty} \gamma^t P_t^\pi(s'|s)$ denotes the discounted state visitation distribution, starting from state $s$ and following policy $\pi$. The following theorem provides an upper bound on the difference between the two losses.

**Theorem 2** (Difference between $\mathcal{L}_\gamma(\pi_\theta, \pi_{\theta_{old}}, s)$ and $\mathcal{L}_k(\pi_\theta, \pi_{\theta_{old}}, s)$)**.** *For a $\gamma$ such that $\gamma^{\frac{1}{1-\gamma}} > \frac{1}{3}$ (e.g., $\gamma \geq 0.85$), any two policies $\pi_{\theta_{old}}$, $\pi_\theta$, and any state $s$, we have:*

$$
|\mathcal{L}_\gamma(\pi_\theta, \pi_{\theta_{old}}, s) - \mathcal{L}_k(\pi_\theta, \pi_{\theta_{old}}, s)| \leq \begin{cases} \mathcal{O}\left(\dfrac{\gamma^k}{1-\gamma}\right), & \text{for } 1 \leq k \leq \dfrac{1}{1-\gamma}, \\ \mathcal{O}(k(1-\gamma)(k-1)+k), & \text{for } k > \dfrac{1}{1-\gamma}. \end{cases}
$$

**Remark 1.** *The function $f(\gamma) = \gamma^{\frac{1}{1-\gamma}}$ is an increasing function for $\gamma \in (0,1)$.*

The bound presented in Theorem 2 reveals a key insight: the upper bound is minimized at $k = \frac{1}{1-\gamma}$, indicating maximal similarity between the two losses at this value. This minimum arises because for $1 \leq k \leq \frac{1}{1-\gamma}$, the bound term $\mathcal{O}\left(\frac{\gamma^k}{1-\gamma}\right)$ decreases as $k$ increases, while for $k > \frac{1}{1-\gamma}$, the bound term $\mathcal{O}(k(1-\gamma)(k-1)+k)$ increases with $k$. This finding formally supports the conventional view that $\frac{1}{1-\gamma}$ acts as the "effective horizon" for the discounted return (Schulman (2016); Sutton & Barto (2018)). Besides, this theorem provides a guideline for choosing $k$ in non-stationary environment.

### 4.3 COMPARISON WITH THE LOSS OF AVERAGE RETURN

We now establish the connection to the average return. We show that the difference between the two loss functions vanishes as the policy converges. Our analysis is grounded in the stationary state distribution $d^\pi$ induced by a policy $\pi$. We begin by defining the overall performance metric as $J_k(\pi) = \mathbb{E}_{s \sim d^\pi}[V_k^\pi(s)]$ and proceed by establishing a performance difference lemma for this metric.

**Lemma 2** (Performance difference under stationary state distribution)**.** *Suppose Assumption 1 holds. For any two policies $\pi$, $\pi'$, the performance difference is given by*

$$
J_k(\pi) - J_k(\pi') = k \cdot \mathbb{E}_{s \sim d^\pi, a \sim \pi}\left[A_k^{\pi'}(s,a)\right] + \sum_s \left(d^\pi(s) - d^{\pi'}(s)\right) V_k^{\pi'}(s) + \mathbb{E}_{s \sim d^\pi}[\Delta(s)],
$$

*where $\Delta(s) = \mathbb{E}_{\pi,P}\left[\sum_{t=1}^{k-1} \mathbb{E}_{\pi',P}[r(s_{t+k-1}, a_{t+k-1})|s_t]\big|s_0 = s\right] - \mathbb{E}_{\pi,P}\left[V_{k-1}^{\pi'}(s_k)\big|s_0 = s\right]$.*

Lemma 2 reveals that the performance difference is driven by the expected advantage and the mismatch between stationary state distributions (notably, by Lemma 10 in the Appendix, $\mathbb{E}_{s \sim d^\pi}[\Delta(s)]$ is also controlled by this distribution mismatch). To bound this mismatch, we must first quantify the distance between stationary distributions. This requires the following definition and assumption.

**Definition 1** (Dobrushin ergodicity coefficient (Rhodius (1997)))**.** *For any given policy $\pi$, the Dobrushin ergodicity coefficient is defined as $\beta(\pi) = \max_{s_1, s_2 \in S} D_{TV}(P^\pi(\cdot|s_1), P^\pi(\cdot|s_2))$.*

**Assumption 2** (Dobrushin ergodicity coefficient bound)**.** *We assume that for any policy $\pi$, its Dobrushin ergodicity coefficient is bounded as $0 < \beta(\pi) < 1$.*

Assumption 2 ensures that the policy-induced transition operator $P^\pi$ is a contraction operator, which is a common assumption for bounding the distance between stationary distributions (Meyn & Tweedie (2012)). This leads to the following result:

**Lemma 3** (Distance between $d^\pi$ and $d^{\pi'}$)**.** *Suppose Assumption 1 and 2 hold. For two policies $\pi$ and $\pi'$ with corresponding stationary distributions $d^\pi$ and $d^{\pi'}$, the total variation distance is bounded by:*

$$
D_{TV}(d^\pi(\cdot), d^{\pi'}(\cdot)) \leq \frac{\max_s D_{TV}(\pi'(\cdot|s), \pi(\cdot|s))}{1 - \beta(\pi')}.
$$

Leveraging this bound, we can formulate the loss and connect its optimization to policy improvement.

**Theorem 3** (Relation between the loss and performance difference under stationary state distribution)**.** *Suppose Assumptions 1 and 2 hold. For a policy $\pi_\theta$ and a previous policy $\pi_{\theta_{old}}$, define the loss as:*

$$
\mathcal{L}_k(\pi_\theta, \pi_{\theta_{old}}) = \mathbb{E}_{s \sim d^{\pi_{\theta_{old}}}, a \sim \pi_{\theta_{old}}}\left[\frac{\pi_\theta(a|s)}{\pi_{\theta_{old}}(a|s)} A_k^{\pi_{\theta_{old}}}(s,a)\right],
$$

*The relationship between policy improvement and this loss is bounded by:*

$$|J_k(\pi_\theta) - J_k(\pi_{\theta_{old}}) - k \cdot \mathcal{L}_k(\pi_\theta, \pi_{\theta_{old}})| \leq 4k \frac{D_{\mathrm{TV}}^{\max}(\pi_{\theta_{old}} \| \pi_\theta)}{1 - \beta(\pi_{\theta_{old}})} \left( \alpha D_{\mathrm{TV}}^{\max}(\pi_{\theta_{old}} \| \pi_\theta) + |R_{\max}| \right),$$

*where $D_{\mathrm{TV}}^{\max}(\pi_{\theta_{old}} \| \pi_\theta) = \max_s D_{\mathrm{TV}}(\pi_{\theta_{old}}(\cdot|s), \pi_\theta(\cdot|s))$ and $\alpha = \max_{s,a} \left| A_k^{\pi_{\theta_{old}}}(s,a) \right|$.*

Theorem 3 thus confirms that the approximation $J_k(\pi_\theta) - J_k(\pi_{\theta_{old}}) \approx k \cdot \mathcal{L}_k(\pi_\theta, \pi_{\theta_{old}})$ holds when the policy update step is small. Next, we establish the connection between our loss $\mathcal{L}_k(\pi_\theta, \pi_{\theta_{old}})$ and the loss for the average return, $\bar{\mathcal{L}}(\pi_\theta, \pi_{\theta_{old}})$. We first define $\bar{\mathcal{L}}(\pi_\theta, \pi_{\theta_{old}})$ (Zhang & Ross (2021)) and mixing time, which is essential for bridging the finite-step sum and the stationary expectation

$$\bar{\mathcal{L}}(\pi_\theta, \pi_{\theta_{old}}) = \mathbb{E}_{s \sim d^{\pi_{\theta_{old}}}, a \sim \pi_{\theta_{old}}} \left[ \frac{\pi_\theta(a|s)}{\pi_{\theta_{old}}(a|s)} \bar{A}^{\pi_{\theta_{old}}}(s,a) \right].$$

**Definition 2** (Mixing time (Levin & Peres (2017))). *For an MDP with transition operator $P^\pi$ under policy $\pi$, the mixing time $T_{\mathrm{mix}}(\xi)$ for a given precision $0 < \xi_k \ll 1$ is defined as:*

$$T_{\mathrm{mix}}(\xi) = \min \left\{ t \in \mathbb{N}^+ \, \middle| \, \max_s D_{\mathrm{TV}}(P_t^\pi(\cdot|s), d^\pi(\cdot|s)) < \xi \right\}.$$

This allows us to bound the difference between the two losses.

**Theorem 4** (Difference between $\mathcal{L}_k(\pi_\theta, \pi_{\theta_{old}})$ and $\bar{\mathcal{L}}(\pi_\theta, \pi_{\theta_{old}})$). *Suppose Assumptions 1 and 2 hold. Given $k$ and $0 < \xi_k \ll 1$ such that $k - 1 \geq T_{\mathrm{mix}}(\xi_k)$, the difference between the losses for policies $\pi_\theta$ and $\pi_{\theta_{old}}$ is bounded by:*

$$\left| \bar{\mathcal{L}}(\pi_\theta, \pi_{\theta_{old}}) - \mathcal{L}_k(\pi_\theta, \pi_{\theta_{old}}) \right| \leq \mathcal{O}\left( \frac{D_{\mathrm{TV}}^{\max}(\pi_{\theta_{old}} \| \pi_\theta) \cdot \xi_k}{1 - \beta(\pi_{\theta_{old}})} \right),$$

*where $D_{\mathrm{TV}}^{\max}(\pi_{\theta_{old}} \| \pi_\theta) = \max_s D_{\mathrm{TV}}(\pi_{\theta_{old}}(\cdot|s), \pi_\theta(\cdot|s))$.*

Theorem 4 establishes that $\mathcal{L}_k(\pi_\theta, \pi_{\theta_{old}})$ approximates $\bar{\mathcal{L}}(\pi_\theta, \pi_{\theta_{old}})$. The quality of this approximation improves as the magnitude of the policy update, i.e., $D_{\mathrm{TV}}^{\max}(\pi_{\theta_{old}} \| \pi_\theta)$, diminishes and as the horizon $k$ increases, which yields a smaller $\xi_k$. This suggests that even though our objective only considers a finite horizon $k$, its unbiased nature is beneficial for performance in long-term stationary tasks.

In summary, our analysis reveals that the $k$-sliding-window return exhibits a dual character. It shares structural similarities with the discounted return, yet its corresponding loss, $\mathcal{L}_k(\pi_\theta, \pi_{\theta_{old}})$, formally converges to the average-return objective $\bar{\mathcal{L}}(\pi_\theta, \pi_{\theta_{old}})$.

Besides, there is a fundamental trade-off in the choice of $k$: On the one hand, a large $k$ aligns the loss for the $k$-sliding-window return more closely with those for the discounted and average return (as shown in Theorems 2 and 4). However, this comes at the cost of weakening the policy improvement guarantee provided by Theorems 1 and 3. On the other hand, a small $k$ ensures a tighter policy improvement bound, but introduces a larger deviation between our loss and the losses of discounted return and average return. We investigate this trade-off empirically in Section 5.

## 5 EXPERIMENT

### 5.1 IMPLEMENTATION DETAIL

In the experiment, we adopt the GAE framework (Schulman et al. (2015)) to estimate the advantage function for the $k$-sliding-window return. A straightforward application of GAE over the finite $k$-step window is defined as:

$$\mathrm{GAE}(\lambda, k, t) = \sum_{i=0}^{k-1} \lambda^i \delta_{t+i}, \text{ where } \delta_{t+i} = r(s_{t+i}, a_{t+i}) + V_k^\pi(s_{t+i+1}) - r(s_{t+k+i}, a_{t+k+i}) - V_k^\pi(s_t).$$

However, directly summing these TD-residuals can introduce high variance because of the term $r(s_{t+k+i}, a_{t+k+i})$ in $\delta_{t+i}$. To strike a better bias-variance trade-off, we propose a modified estimator:

$$\widehat{\mathrm{GAE}}(\lambda, k, t) = \sum_{i=0}^{k-1} \lambda^i \widehat{\delta}_{t+i} - \lambda^{k-1} V_k^\pi(s_{t+k}), \text{ where } \widehat{\delta}_{t+i} = r(s_{t+i}, a_{t+i}) + V_k^\pi(s_{t+i+1}) - V_k^\pi(s_t).$$

While this formulation introduces a bias for $\lambda \neq 1$, it exhibits a lower variance in practice, which results in more stable training. Notably, when $\lambda = 1$, the bias vanishes and our estimator correctly recovers the advantage function:

$$\mathbb{E}\left[\text{GAE}(1, k, t)\right] = \mathbb{E}\left[\widehat{\text{GAE}}(1, k, t)\right] = \mathbb{E}\left[\sum_{i=0}^{k-1} r(s_{t+i}, a_{t+i})\middle| s_t, a_t\right] - V_k^{\pi}(s_t).$$

This indicates that the estimator's bias is well-controlled by selecting $\lambda$ close to 1.

For the value function update, we employ the smooth $L^1$ loss, which is more robust to outlier value estimates than the standard $L^2$ loss. For continuous control tasks, we model the policy using a Beta distribution. This approach is chosen over the common Tanh-squashed Gaussian policy to improve numerical stability.

## 5.2 MAIN RESULT

We evaluate the performance of the proposed $k$SW-PPO algorithm against two baselines: the discounted-return PPO and an average-return PPO. The comparison is conducted on a suite of MuJoCo continuous control tasks, categorized into stationary environments (*Hopper, Walker2d, HalfCheetah, Swimmer*) and non-stationary environments (*Reacher, Pusher*). Their state visitation patterns are given in Appendix H. While we discussed several enhancements to discounted return methods in the related work, we use the vanilla PPO (Schulman et al. (2017)) as our discounted-return baseline for a fair comparison. This is because the method by Tang et al. (2021) lacks a public implementation, and the approach by Gao et al. (2022) is designed to prevent periodic behaviors, making it unsuitable for stationary environments where such periodicity is desirable.

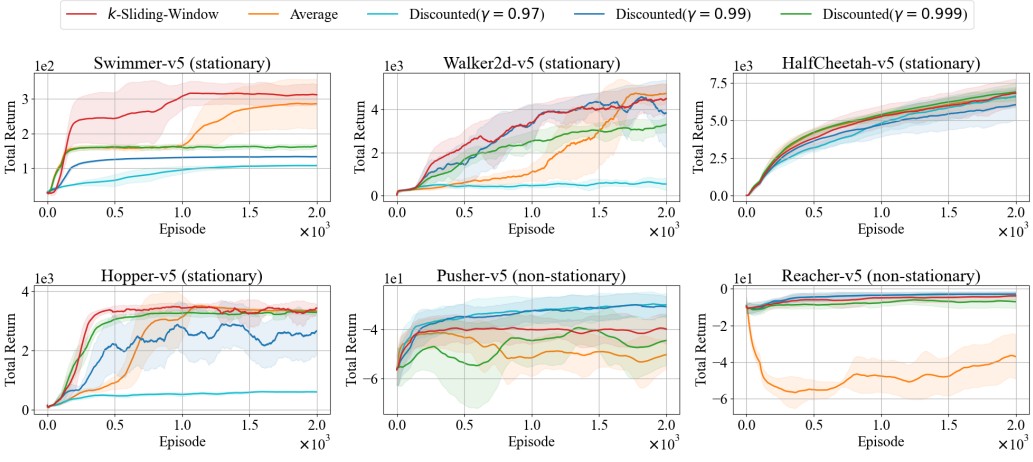

Figure 2: Performance comparison of $k$SW-PPO, average-return PPO, and discounted-return PPO. The learning curves show the average episodic return throughout training, evaluated across 5 random seeds. Solid lines indicate the mean return smoothed over a 100-episode window, and the shaded regions represent the standard deviation.

Experimental setup details are provided in Appendix I. For our method, the window size $k$ was set to 50 for *Swimmer, Walker2d*, and *Hopper*; 30 for *HalfCheetah*; and 10 for *Reacher* and *Pusher*.

The results in Figure 2 validate our central claim: $k$SW-PPO serves as a robust and unified objective, which performs competitively across both stationary and non-stationary environments. In the stationary tasks, $k$SW-PPO consistently performs on par with the specialized average-return PPO. In *Swimmer* and *Walker2d*, both methods outperform the discounted-return PPO. In *HalfCheetah* and *Hopper*, their performance is competitive with the best-performing discounted baseline ($\gamma = 0.999$). Conversely, in the non-stationary tasks, $k$SW-PPO's performance aligns with that of the discounted-return PPO. In *Reacher*, both substantially outperform the failing average-return PPO. While in *Pusher*, $k$SW-PPO does not fully match the top performance of its discounted counterpart, it still maintains a considerable performance margin over the average-return variant, which struggles in this

setting. Collectively, these results demonstrate that $k$SW-PPO successfully inherits the strengths of both criteria, eliminating the need for an a priori choice of objective.

### 5.3 ABLATION STUDY

The window size, $k$, is a key hyperparameter that governs the behavior of $k$SW-PPO. As established in our theoretical analysis (Section 4), its choice embodies a fundamental trade-off. This sensitivity to a core hyperparameter is not a drawback but is analogous to the critical role of the discount factor.

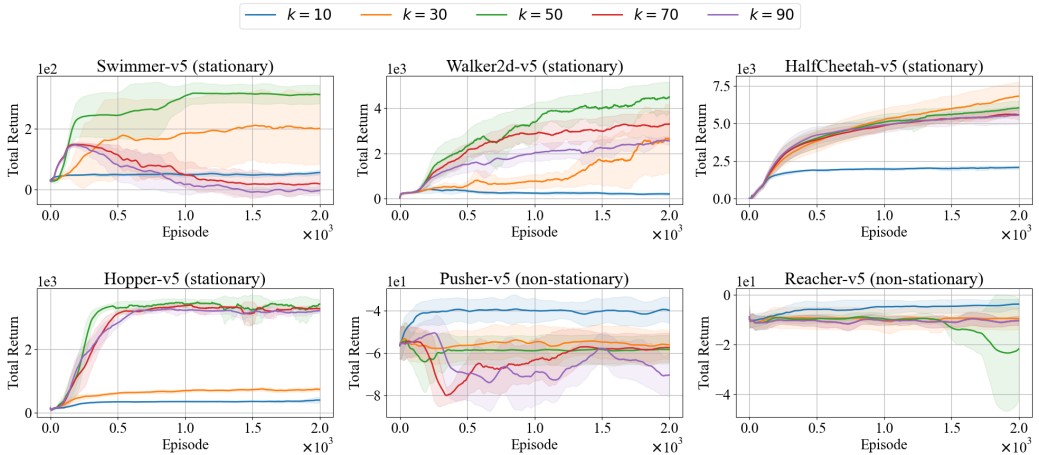

Figure 3: Performance comparison between different window size, $k$, for $k$SW-PPO. The learning curves show the average episodic return throughout training, evaluated across 5 random seeds. Solid lines indicate the mean return smoothed over a 100-episode window, and the shaded regions represent the standard deviation.

To empirically investigate this trade-off and the sensitivity to $k$, we conducted an ablation study across our suite of MuJoCo environments. The performance for a range of $k$ values, $\{10, 30, 50, 70, 90\}$, is presented in Figure 3. While performance is indeed sensitive to $k$, the results reveal a remarkable consistency across environments. Notably, $k = 50$ yields optimal or near-optimal performance in all stationary environments. Similarly, a smaller $k = 10$ is preferred for both non-stationary environments. This finding is of significant practical value. It suggests that the optimal choice of $k$ is not arbitrarily specific to each task but exhibits cross-environment consistency. This underscores the inherent robustness and generality of the $k$-sliding-window return itself.

## 6 CONCLUSION

In this work, we addressed the long-standing dichotomy between discounted and average returns that has constrained the generality of DRL algorithms. We introduced the $k$-sliding-window return, a novel objective designed to bridge these two criteria. This concept was instantiated through a practical on-policy algorithm, $k$SW-PPO. Our theoretical analysis reveals the dual nature of this objective: we proved that the loss of $k$SW-PPO converges to that of the average-return, while also showing it maintains a bounded bias relative to the loss of the discounted-return. Our claims were validated through a series of experiments on MuJoCo continuous control tasks, where $k$SW-PPO demonstrated its robustness. It performed competitively with average-return PPO in stationary settings and matched the performance of discounted-return PPO in non-stationary ones. Ultimately, our results establish the $k$-sliding-window return as a unified objective. This eliminates the need for practitioners to make an a priori choice between the two criteria, effectively addressing the initial problem.

## REPRODUCIBILITY STATEMENT

We provide the source code and configuration for the key experiments including instructions on how to generate data and train the models in link `https://anonymous.4open.science/r/kSW-PPO-1E7C`. We thoroughly checked the implementation. All proofs are stated in the appendix with explanations and underlying assumptions.

## ETHICS STATEMENT

The authors of this paper have read and agree to adhere to the ICLR Code of Ethics.

Our work is theoretical and algorithmic, introducing a new objective for RL and a corresponding on-policy algorithm. The experiments are conducted in standard simulated environments commonly used for RL research (i.e., MuJoCo), and do not involve any human subjects or personally identifiable or sensitive data.

However, we acknowledge that as with any advancement in RL, there are potential downstream societal impacts to consider. The behavior of an RL agent is fundamentally shaped by its reward function and training environment. If our algorithm were to be deployed in real-world applications, it could be susceptible to learning unintended or biased behaviors if the reward signal is not carefully designed. Furthermore, RL as a field has dual-use potential; powerful decision-making agents could be applied to applications with negative consequences if not developed and deployed with care.

While our foundational contribution does not in itself introduce new ethical risks, we urge practitioners who build upon our work to consider the specific context of their application, to rigorously audit for potential biases, and to ensure that reward functions align with safe and socially beneficial outcomes.

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

## A    PROOF OF LEMMA 1

**Lemma 1** (Performance difference). *For any two policies $\pi$ and $\pi'$, and any initial state $s$, the following holds:*

$$V_k^\pi(s) - V_k^{\pi'}(s) = k \cdot \mathbb{E}_{s' \sim d_{k,s}^\pi, a \sim \pi} \left[ A_k^{\pi'}(s', a) \right] + \Delta(s).$$

*where $\Delta(s) = \mathbb{E}_{\pi, P} \left[ \sum_{t=1}^{k-1} \mathbb{E}_{\pi', P}[r(s_{t+k-1}, a_{t+k-1})|s_t] \big| s_0 = s \right] - \mathbb{E}_{\pi, P} \left[ V_{k-1}^{\pi'}(s_k) \big| s_0 = s \right].$*

*Proof.* From the definition of $V_k^\pi(s)$, we have

$$V_k^\pi(s) - V_k^{\pi'}(s)$$

$$= \mathbb{E}_{\pi, P} \left[ \sum_{t=0}^{k-1} r(s_t, a_t) \bigg| s_0 = s \right] - V_k^{\pi'}(s)$$

$$= \mathbb{E}_{\pi, P} \left[ \sum_{t=0}^{k-1} \left( r(s_t, a_t) + V_k^{\pi'}(s_t) - V_k^{\pi'}(s_t) \right) \bigg| s_0 = s \right] - V_k^{\pi'}(s)$$

$$= \mathbb{E}_{\pi, P} \left[ \sum_{t=1}^{k-1} \left( r(s_t, a_t) + V_{k-1}^{\pi'}(s_t) + \mathbb{E}_{\pi', P} \left[ r(s_{t+k-1}, a_{t+k-1})|s_t \right] - V_k^{\pi'}(s_t) \right) \right.$$

$$\left. + \left( r(s_0, a_0) + V_k^{\pi'}(s_0) - V_k^{\pi'}(s_0) \right) \bigg| s_0 = s \right] - V_k^{\pi'}(s)$$

$$\overset{(*)}{=} \mathbb{E}_{\pi, P} \left[ \sum_{t=0}^{k-1} \left( r(s_t, a_t) + V_{k-1}^{\pi'}(s_{t+1}) - V_k^{\pi'}(s_t) \right) \right.$$

$$\left. + \left( \sum_{t=1}^{k-1} \mathbb{E}_{\pi', P}[r(s_{t+k-1}, a_{t+k-1})|s_t] - V_{k-1}^{\pi'}(s_k) \right) + V_k^{\pi'}(s_0) \bigg| s_0 = s \right] - V_k^{\pi'}(s)$$

$$\overset{(**)}{=} \mathbb{E}_{\pi, P} \left[ \sum_{t=0}^{k-1} \left( r(s_t, a_t) + \mathbb{E}_{s_{t+1} \sim P(\cdot|s_t, a_t)} \left[ V_{k-1}^{\pi'}(s_{t+1}) \right] - V_k^{\pi'}(s_t) \right) \bigg| s_0 = s \right] + \Delta(s)$$

$$= k \cdot \mathbb{E}_{s' \sim d_{k,s}^\pi, a \sim \pi} \left[ A_k^{\pi'}(s', a) \right] + \Delta(s),$$

where we define

$$\Delta(s) = \mathbb{E}_{\pi, P} \left[ \sum_{t=1}^{k-1} \mathbb{E}_{\pi', P}[r(s_{t+k-1}, a_{t+k-1})|s_t] \bigg| s_0 = s \right] - \mathbb{E}_{\pi, P} \left[ V_{k-1}^{\pi'}(s_k) \bigg| s_0 = s \right].$$

The equation $(*)$ is constructed through rearrange the terms and add $V_{k-1}^\pi(s_k) - V_{k-1}^\pi(s_k)$. The equation $(**)$ holds because of the tower property of expectation. The proof is complete. $\qquad\square$

## B    PROOF OF THEOREM 1

The proof of Theorem 1 proceeds in several steps. The core of the argument is to bound the difference between the $k$-step state visitation distributions, $d_{k,s}^\pi(\cdot)$ and $d_{k,s}^{\pi'}(\cdot)$. To achieve this, we first need a bound on the distance between their respective $t$-step transition probabilities, which is established in Lemma 4.

**Lemma 4** (Total variation distance between $P_t^\pi$ and $P_t^{\pi'}$). *Given a transition probability $P$, two policies $\pi$ and $\pi'$, and a step size $t \geq 1$, the total variation distance between $P_t^\pi$ and $P_t^{\pi'}$ is bounded as*

$$D_{\text{TV}}(P_t^\pi(\cdot|s), P_t^{\pi'}(\cdot|s)) \leq t \max_s D_{\text{TV}}(\pi(\cdot|s), \pi'(s)).$$

*Proof.* From the definition of $D_{\mathrm{TV}}(P_t^\pi(\cdot|s), P_t^{\pi'}(\cdot|s))$, we have

$$D_{\mathrm{TV}}(P_t^\pi(\cdot|s), P_t^{\pi'}(\cdot|s)) = \frac{1}{2}\sum_{s'}\left|P_t^\pi(s'|s) - P_t^{\pi'}(s'|s)\right|.$$

Then for the difference between two distribution probability under policy $\pi, \pi'$, we have

$$\left|P_t^\pi(s'|s) - P_t^{\pi'}(s'|s)\right|$$

$$= \left|\sum_{s''} P_{t-1}^\pi(s''|s)P^\pi(s'|s'') - \sum_{s''} P_{t-1}^{\pi'}(s''|s)P^{\pi'}(s'|s'')\right|$$

$$\leq \sum_{s''}\left|P_{t-1}^\pi(s''|s)P^\pi(s'|s'') - P_{t-1}^{\pi'}(s''|s)P^{\pi'}(s'|s'')\right|$$

$$= \sum_{s''}\left|P_{t-1}^\pi(s''|s)P^\pi(s'|s'') - P_{t-1}^{\pi'}(s''|s)P^\pi(s'|s'')\right.$$

$$\left. + P_{t-1}^{\pi'}(s''|s)P^\pi(s'|s'') - P_{t-1}^{\pi'}(s''|s)P^{\pi'}(s'|s'')\right|$$

$$= \sum_{s''}\left|\left(P_{t-1}^\pi(s''|s) - P_{t-1}^{\pi'}(s''|s)\right)P^\pi(s'|s'') + P_{t-1}^{\pi'}(s''|s)\left(P^\pi(s'|s'') - P^{\pi'}(s'|s'')\right)\right|$$

$$\leq \sum_{s''}\left|P_{t-1}^\pi(s''|s) - P_{t-1}^{\pi'}(s''|s)\right|P^\pi(s'|s'') + \sum_{s''}P_{t-1}^{\pi'}(s''|s)\left|P^\pi(s'|s'') - P^{\pi'}(s'|s'')\right|.$$

Because for any $s''$, $\sum_{s'} P^\pi(s'|s'') = 1$, we can have the following recursion inequality

$$\sum_{s'}\left|P_t^\pi(s'|s) - P_t^{\pi'}(s'|s)\right| \leq \sum_{s''}\left|P_{t-1}^\pi(s''|s) - P_{t-1}^{\pi'}(s''|s)\right| +$$

$$\sum_{s''}P_{t-1}^{\pi'}(s''|s)\sum_{s'}\left|P^\pi(s'|s'') - P^{\pi'}(s'|s'')\right|.$$

Expend the first term in the RHS of the above inequality based on $t$, we have

$$\sum_{s'}\left|P_t^\pi(s'|s) - P_t^{\pi'}(s'|s)\right| \leq \sum_{i=0}^{t-1}\sum_{s''}P_i^{\pi'}(s''|s)\sum_{s'}\left|P^\pi(s'|s'') - P^{\pi'}(s'|s'')\right|.$$

The absolute term in the RHS of the above inequality can be bounded as

$$\sum_{s'}\left|P^\pi(s'|s'') - P^{\pi'}(s'|s'')\right| = \sum_{s'}\sum_{a}P(s'|s'',a)|\pi(a|s'') - \pi'(a|s'')|$$

$$= \sum_{a}\sum_{s'}P(s'|s'',a)|\pi'(a|s'') - \pi(a|s'')|$$

$$= \sum_{a}|\pi'(a|s'') - \pi(a|s'')|$$

$$= 2D_{\mathrm{TV}}(\pi'(\cdot|s''), \pi(\cdot|s'')).$$

So we can have

$$\sum_{s'}\left|P_t^\pi(s'|s) - P_t^{\pi'}(s'|s)\right| \leq 2\sum_{i=0}^{t-1}\sum_{s''}P_i^{\pi'}(s''|s)D_{\mathrm{TV}}(\pi'(\cdot|s''), \pi(\cdot|s''))$$

$$\leq 2\sum_{i=0}^{t-1}\sum_{s''}P_i^{\pi'}(s''|s)\max_s D_{\mathrm{TV}}(\pi'(\cdot|s), \pi'(\cdot|s))$$

$$= 2t\max_s D_{\mathrm{TV}}(\pi'(\cdot|s), \pi(\cdot|s)).$$

So, finally we have

$$D_{\mathrm{TV}}(P_t^\pi(\cdot|s), P_t^{\pi'}(\cdot|s)) \leq t\max_s D_{\mathrm{TV}}(\pi'(\cdot|s), \pi(\cdot|s)),$$

which completes the proof. $\qquad\square$

Lemma 5 gives a basic property of the TV distance.

**Lemma 5** (Non-expansion property of total variation distance). *Given a transition probability $P_t^\pi$ corresponding to policy $\pi$ with step size $t \geq 1$, and two probability distribution $\mu$ and $\nu$, there is*

$$D_{\mathrm{TV}}(P_t^\pi \mu(\cdot), P_t^\pi \nu(\cdot)) \leq D_{\mathrm{TV}}(\mu(\cdot), \nu(\cdot)).$$

*Proof.* From the definition, we have

$$D_{\mathrm{TV}}(P_t^\pi \mu(\cdot), P_t^\pi \nu(\cdot)) = \frac{1}{2} \sum_{s'} \left| \sum_s P_t^\pi(s'|s)\mu(s) - \sum_s P_t^\pi(s'|s)\nu(s) \right|$$

$$= \frac{1}{2} \sum_{s'} \left| \sum_s P_t^\pi(s'|s)(\mu(s) - \nu(s)) \right|$$

$$\leq \frac{1}{2} \sum_{s'} \sum_s P_t^\pi(s'|s) |\mu(s) - \nu(s)|$$

$$= \frac{1}{2} \left( \sum_{s'} P_t^\pi(s'|s) \right) \sum_s |\mu(s) - \nu(s)|$$

$$= \frac{1}{2} \sum_s |\mu(s) - \nu(s)| = D_{\mathrm{TV}}(\mu(\cdot), \nu(\cdot)).$$

The proof is completed. $\qquad\square$

By combining Lemma 4 with a fundamental non-expansion property of the TV distance, we are able to prove Lemma 6, which bounds the extra term $\Delta$, and Lemma 7, which provides the required bound on the difference between the state visitation distributions.

**Lemma 6** (Bound of $\Delta$). *Given two policy $\pi$ and $\pi'$, and a initial state $s$, define $\Delta$ as*

$$\Delta(s) = \mathbb{E}_{\pi,P} \left[ \sum_{t=1}^{k-1} \mathbb{E}_{\pi',P}[r(s_{t+k-1}, a_{t+k-1})|s_t] \middle| s_0 = s \right] - \mathbb{E}_{\pi,P} \left[ V_{k-1}^{\pi'}(s_k) \middle| s_0 = s \right],$$

*there are*

$$\Delta(s) \leq k(k-1)|R_{\max}| \max_s D_{\mathrm{TV}}(\pi(\cdot|s), \pi'(\cdot|s)).$$

*Proof.* From the definition, we can have

$$\mathbb{E}_{\pi,P} \left[ V_{k-1}^{\pi'}(s_k) \middle| s_0 = s \right] = \sum_{s_k} P_k^\pi(s_k|s_0) \sum_{h=k}^{2k-2} \sum_{s_h} P_{h-k}^{\pi'}(s_h|s_k) \sum_a \pi'(a|s_h)r(s_h, a)$$

$$= \sum_{h=k}^{2k-2} \sum_{s_h} \sum_{s_k} P_k^\pi(s_k|s_0) P_{h-k}^{\pi'}(s_h|s_k) \sum_a \pi'(a|s_h)r(s_h, a),$$

and

$$\mathbb{E}_{\pi,P} \left[ \sum_{t=1}^{k-1} \mathbb{E}_{\pi',P}[r(s_{t+k-1}, a_{t+k-1})|s_t] \right]$$

$$= \sum_{h=k}^{2k-2} \sum_{s_h} \sum_{s_{h-k+1}} P_{h-k+1}^\pi(s_{h-k+1}|s_0) \sum_{s_h} P_{k-1}^{\pi'}(s_h|s_{h-k+1}) \sum_a \pi'(a|s_h)r(s_h, a).$$

So we have

$$\Delta = \sum_{h=k}^{2k-2} \sum_{s_h} \left( \sum_{s_{h-k+1}} P_{h-k+1}^\pi(s_{h-k+1}|s_0)P_{k-1}^{\pi'}(s_h|s_{h-k+1}) - \sum_{s_k} P_k^\pi(s_k|s_0)P_{h-k}^{\pi'}(s_h|s_k) \right)$$

$$\times \left( \sum_a \pi'(a|s_h) r(s_h, a) \right)$$

$$\leq \sum_{h=k}^{2k-2} \sum_{s_h} \left| \sum_{s_{h-k+1}} P_{h-k+1}^\pi(s_{h-k+1}|s_0) P_{k-1}^{\pi'}(s_h|s_{h-k+1}) - \sum_{s_k} P_k^\pi(s_k|s_0) P_{h-k}^{\pi'}(s_h|s_k) \right| |R_{\max}|.$$

Given any $k \leq h \leq 2k-2$, we define $i = h - k$, then we have

$$\sum_{s_h} \left| \sum_{s_{h-k+1}} P_{h-k+1}^\pi(s_{h-k+1}|s_0) P_{k-1}^{\pi'}(s_h|s_{h-k+1}) - \sum_{s_k} P_k^\pi(s_k|s_0) P_{h-k}^{\pi'}(s_h|s_k) \right|$$

$$= \sum_{s_{k+i}} \left| \sum_{s_{i+1}} P_{i+1}^\pi(s_{i+1}|s_0) P_{k-1}^{\pi'}(s_{k+i}|s_{i+1}) - \sum_{s_k} P_k^\pi(s_k|s_0) P_i^{\pi'}(s_{k+i}|s_k) \right|$$

$$= \sum_{s_{k+i}} \left| \sum_{s_{i+1}} P_{i+1}^\pi(s_{i+1}|s_0) \sum_{s_k} \left( P_{k-i-1}^{\pi'}(s_k|s_{i+1}) - P_{k-i-1}^{\pi}(s_k|s_{i+1}) \right) P_i^{\pi'}(s_{k+i}|s_k) \right|$$

$$\overset{(*)}{\leq} \sum_{s_{k+i}} \sum_{s_{i+1}} P_{i+1}^\pi(s_{i+1}|s_0) \left| \sum_{s_k} \left( P_{k-i-1}^{\pi'}(s_k|s_{i+1}) - P_{k-i-1}^{\pi}(s_k|s_{i+1}) \right) P_i^{\pi'}(s_{k+i}|s_k) \right|$$

$$= \sum_{s_{i+1}} P_{i+1}^\pi(s_{i+1}|s_0) \sum_{s_{k+i}} \left| \sum_{s_k} \left( P_{k-i-1}^{\pi'}(s_k|s_{i+1}) - P_{k-i-1}^{\pi}(s_k|s_{i+1}) \right) P_i^{\pi'}(s_{k+i}|s_k) \right|$$

$$\overset{(**)}{\leq} \sum_{s_{i+1}} P_{i+1}^\pi(s_{i+1}|s_0) \left( \sum_{s_k} \left| P_{k-i-1}^{\pi'}(s_k|s_{i+1}) - P_{k-i-1}^{\pi}(s_k|s_{i+1}) \right| \right)$$

$$\overset{(***)}{\leq} \sum_{s_{i+1}} P_{i+1}^\pi(s_{i+1}|s_0) \left( 2(k-i-1) \max_s D_{\mathrm{TV}}(\pi'(\cdot|s), \pi(\cdot|s)) \right)$$

$$= 2(k-i-1) \max_s D_{\mathrm{TV}}(\pi'(\cdot|s), \pi(\cdot|s)),$$

where inequality $(*)$ holds because of Jenson's inequality, inequality $(**)$ holds because of the non-expansion property of total variation distance from Lemma 5, and inequality $(***)$ holds because of Lemma 4. So we now have

$$\Delta \leq |R_{\max}| \sum_{i=0}^{k-2} 2(k-i-1) \max_s D_{\mathrm{TV}}(\pi'(\cdot|s), \pi(\cdot|s))$$

$$= k(k-1)|R_{\max}| \max_s D_{\mathrm{TV}}(\pi'(\cdot|s), \pi(\cdot|s)).$$

This completes the proof. $\qquad\square$

**Lemma 7** (Difference between two distribution of policies). *Given two distribution $d_{k,s}^\pi$, $d_{k,s}^{\pi'}$ respect to two policies $\pi$, $\pi'$ and initial state $s$, the total variation distance between two distribution satisfies*

$$\sum_{s'} \left| d_{k,s}^\pi(s') - d_{k,s}^{\pi'}(s') \right| \leq (k-1) \max_s D_{\mathrm{TV}}(\pi'(\cdot|s), \pi(\cdot|s)).$$

*Proof.* From the definition of $d_{k,s}^\pi(s')$, we have

$$\sum_{s'} \left| d_{k,s}^\pi(s') - d_{k,s}^{\pi'}(s') \right| = \frac{1}{k} \sum_{s'} \left| \sum_{t=0}^{k-1} \left( P_t^\pi(s'|s) - P_t^{\pi'}(s'|s) \right) \right|$$

$$\leq \frac{1}{k} \sum_{s'} \sum_{t=0}^{k-1} \left| P_t^\pi(s'|s) - P_t^{\pi'}(s'|s) \right| \qquad (1)$$

$$= \frac{2}{k} \sum_{t=0}^{k-1} D_{\mathrm{TV}}(P_t^\pi(\cdot|s), P_t^{\pi'}(\cdot|s)).$$

Using Lemma 4, we further have

$$\sum_{s'} |d_{k,s}^{\pi}(s') - d_{k,s}^{\pi'}(s')| \leq \frac{2}{k} \sum_{t=0}^{k-1} t \max_s D_{\mathrm{TV}}(\pi'(\cdot|s), \pi(\cdot|s))$$
$$= (k-1) \max_s D_{\mathrm{TV}}(\pi'(\cdot|s), \pi(\cdot|s)).$$

The proof is complete. $\qquad\square$

Finally, armed with the results from Lemma 6 and Lemma 7, the proof of Theorem 1 is completed.

**Theorem 1** (Relation between loss and performance difference). *For a policy $\pi_\theta$ and a previous policy $\pi_{\theta_{old}}$, define the loss as:*

$$\mathcal{L}_k(\pi_\theta, \pi_{\theta_{old}}, s) = \mathbb{E}_{s' \sim d_{k,s}^{\pi_{\theta_{old}}}, a \sim \pi_{\theta_{old}}} \left[ \frac{\pi_\theta(a|s')}{\pi_{\theta_{old}}(a|s')} A_k^{\pi_{\theta_{old}}}(s', a) \right].$$

*The relationship between policy improvement and this loss is bounded by:*

$$\left| V_k^{\pi_\theta}(s) - V_k^{\pi_{\theta_{old}}}(s) - k \cdot \mathcal{L}_k(\pi_\theta, \pi_{\theta_{old}}, s) \right| \leq k(k-1) D_{\mathrm{TV}}^{\max}(\pi_{\theta_{old}}\|\pi_\theta)(4\alpha D_{\mathrm{TV}}^{\max}(\pi_{\theta_{old}}\|\pi_\theta)+|R_{\max}|),$$

*where $D_{\mathrm{TV}}^{\max}(\pi_{\theta_{old}}\|\pi_\theta) = \max_s D_{\mathrm{TV}}(\pi_{\theta_{old}}(\cdot|s), \pi_\theta(\cdot|s))$ and $\alpha = \max_{s,a} \left| A_k^{\pi_{\theta_{old}}}(s,a) \right|$.*

*Proof.* For the convenience of discussion, we define

$$\xi(s') := \mathbb{E}_{a \sim \pi_{\theta_{old}}(\cdot|s')} \left[ \frac{\pi_\theta(a|s')}{\pi_{\theta_{old}}(a|s')} A_k^{\pi_{\theta_{old}}}(s', a) \right] = \mathbb{E}_{a \sim \pi_\theta(\cdot|s')} \left[ A_k^{\pi_{\theta_{old}}}(s', a) \right].$$

Because $\mathbb{E}_{a \sim \pi_{\theta_{old}}(\cdot|s')} [A^{\pi_{\theta_{old}}}(s', a)] = 0$, then $|\xi(s')|$ satisfies

$$|\xi(s')| = \left| \sum_a \pi_\theta(a|s') A_k^{\pi_{\theta_{old}}}(s', a) - \sum_a \pi_{\theta_{old}}(a|s') A_k^{\pi_{\theta_{old}}}(s', a) \right|$$
$$= \left| \sum_a \left( \pi_\theta(a|s') - \pi_{\theta_{old}}(a|s') \right) A_k^{\pi_{\theta_{old}}}(s', a) \right|$$
$$\leq 2 \max_s D_{\mathrm{TV}}(\pi_{\theta_{old}}(\cdot|s), \pi_\theta(\cdot|s)) \max_{s,a} |A_k^{\pi_{\theta_{old}}}(s, a)|.$$

From Lemma 1, we can have

$$V_k^{\pi_\theta}(s) - V_k^{\pi_{\theta_{old}}}(s) = k \cdot \mathbb{E}_{s' \sim d_{k,s}^{\pi_\theta}} [\xi(s')] + \Delta, \quad \mathcal{L}_k(\pi_\theta; \pi_{\theta_{old}}, s) = \mathbb{E}_{s' \sim d_{k,s}^{\pi_{\theta_{old}}}} [\xi(s')].$$

Combining with Lemma 6 and Lemma 7, we can have

$$\left| V_k^{\pi_\theta}(s) - V_k^{\pi_{\theta_{old}}}(s) - k \cdot \mathcal{L}_k(\pi_\theta, \pi_{\theta_{old}}, s) \right|$$
$$= k \cdot \left| \sum_{s'} \left( d_{k,s}^{\pi_\theta}(s') - d_{k,s}^{\pi_{\theta_{old}}}(s') \right) \xi(s') + \Delta \right|$$
$$\leq k \sum_{s'} \left| d_{k,s}^{\pi_\theta}(s') - d_{k,s}^{\pi_{\theta_{old}}}(s') \right| |\xi(s')| + |\Delta|$$
$$\leq 4\alpha k(k-1) \left( D_{\mathrm{TV}}^{\max}(\pi_{\theta_{old}}\|\pi_\theta) \right)^2 + k(k-1) D_{\mathrm{TV}}^{\max}(\pi_{\theta_{old}}\|\pi_\theta)|R_{\max}|$$
$$= k(k-1) D_{\mathrm{TV}}^{\max}(\pi_{\theta_{old}}\|\pi_\theta)(4\alpha D_{\mathrm{TV}}^{\max}(\pi_{\theta_{old}}\|\pi_\theta) + |R_{\max}|),$$

where $D_{\mathrm{TV}}^{\max}(\pi_{\theta_{old}}\|\pi_\theta) = \max_s D_{\mathrm{TV}}(\pi_{\theta_{old}}(\cdot|s), \pi_\theta(\cdot|s))$ and $\alpha = \max_{s,a} \left| A_k^{\pi_{\theta_{old}}}(s, a) \right|$. This completes the proof. $\qquad\square$

## C  PROOF OF THEOREM 2

**Theorem 2** (Difference between $\mathcal{L}_\gamma(\pi_\theta, \pi_{\theta_{\text{old}}}, s)$ and $\mathcal{L}_k(\pi_\theta, \pi_{\theta_{\text{old}}}, s)$). *For a $\gamma$ such that $\gamma^{\frac{1}{1-\gamma}} > \frac{1}{3}$ (e.g., $\gamma \geq 0.85$), any two policies $\pi_{\theta_{old}}$, $\pi_\theta$, and any state $s$, we have:*

$$
|\mathcal{L}_\gamma(\pi_\theta, \pi_{\theta_{\text{old}}}, s) - \mathcal{L}_k(\pi_\theta, \pi_{\theta_{\text{old}}}, s)| \leq
\begin{cases}
\mathcal{O}\left(\dfrac{\gamma^k}{1-\gamma}\right), & \text{for } 1 \leq k \leq \dfrac{1}{1-\gamma}, \\[3mm]
\mathcal{O}(k(1-\gamma)(k-1) + k), & \text{for } k > \dfrac{1}{1-\gamma}.
\end{cases}
$$

*Proof.* The difference between two losss is

$$
\begin{aligned}
&\mathcal{L}_\gamma(\pi_\theta, \pi_{\theta_{\text{old}}}, s) - \mathcal{L}_k(\pi_\theta, \pi_{\theta_{\text{old}}}, s) \\
&= \mathbb{E}_{s' \sim d_s^{\pi_{\theta_{\text{old}}}}, a \sim \pi_{\theta_{\text{old}}}} \left[ \frac{\pi_\theta(a|s')}{\pi_{\theta_{\text{old}}}(a|s')} A_\gamma^{\pi_{\theta_{\text{old}}}}(s', a) \right] - \mathbb{E}_{s' \sim d_{k,s}^{\pi_{\theta_{\text{old}}}}, a \sim \pi_{\theta_{\text{old}}}} \left[ \frac{\pi_\theta(a|s')}{\pi_{\theta_{\text{old}}}(a|s')} A_k^{\pi_{\theta_{\text{old}}}}(s', a) \right] \\
&= \sum_{s'} d_s^{\pi_{\theta_{\text{old}}}}(s') \sum_a \pi_\theta(a|s') A_\gamma^{\pi_{\theta_{\text{old}}}}(s', a) - \sum_{s'} d_{k,s}^{\pi_{\theta_{\text{old}}}}(s') \sum_a \pi_\theta(a|s') A_k^{\pi_{\theta_{\text{old}}}}(s', a) \\
&= \sum_{s'} \left( d_s^{\pi_{\theta_{\text{old}}}}(s') - d_{k,s}^{\pi_{\theta_{\text{old}}}}(s') \right) \sum_a \pi_\theta(a|s') A_\gamma^{\pi_{\theta_{\text{old}}}}(s', a) \\
&\quad + \sum_{s'} d_{k,s}^{\pi_{\theta_{\text{old}}}}(s') \sum_a \pi_\theta(a|s') \left( A_\gamma^{\pi_{\theta_{\text{old}}}}(s', a) - A_k^{\pi_{\theta_{\text{old}}}}(s', a) \right).
\end{aligned}
\tag{2}
$$

For the first term in the RHS of the equation (2)

$$
\begin{aligned}
&\left| \sum_{s'} \left( d_s^{\pi_{\theta_{\text{old}}}}(s') - d_{k,s}^{\pi_{\theta_{\text{old}}}}(s') \right) \sum_a \pi_\theta(a|s') A_\gamma^{\pi_{\theta_{\text{old}}}}(s', a) \right| \\
&\leq \left| \sum_{s'} \left( d_s^{\pi_{\theta_{\text{old}}}}(s') - d_{k,s}^{\pi_{\theta_{\text{old}}}}(s') \right) \right| \max_{s,a} \left| A_\gamma^{\pi_{\theta_{\text{old}}}}(s, a) \right| \leq \sum_{s'} \left| d_s^{\pi_{\theta_{\text{old}}}}(s') - d_{k,s}^{\pi_{\theta_{\text{old}}}}(s') \right| \frac{2|R_{\max}|}{1-\gamma}.
\end{aligned}
$$

We further have

$$
\begin{aligned}
\sum_{s'} \left| d_s^{\pi_{\theta_{\text{old}}}}(s') - d_{k,s}^{\pi_{\theta_{\text{old}}}}(s') \right| &= \sum_{s'} \left| (1-\gamma) \sum_{t=0}^\infty \gamma^t P_t^{\pi_{\theta_{\text{old}}}}(s'|s) - \frac{1}{k} \sum_{t=0}^{k-1} P_t^{\pi_{\theta_{\text{old}}}}(s'|s) \right| \\
&= \sum_{s'} \left| \sum_{t=0}^\infty \left( (1-\gamma)\gamma^t - \frac{I(0 \leq t < k)}{k} \right) P_t^{\pi_{\theta_{\text{old}}}}(s'|s) \right| \\
&\leq \sum_{s'} \sum_{t=0}^\infty \left| (1-\gamma)\gamma^t - \frac{I(0 \leq t < k)}{k} \right| P_t^{\pi_{\theta_{\text{old}}}}(s'|s) \\
&= \sum_{t=0}^\infty \left| (1-\gamma)\gamma^t - \frac{I(0 \leq t < k)}{k} \right| \left( \sum_{s'} P_t^{\pi_{\theta_{\text{old}}}}(s'|s) \right) \\
&= \sum_{t=0}^\infty \left| (1-\gamma)\gamma^t - \frac{I(0 \leq t < k)}{k} \right|,
\end{aligned}
$$

where $I(0 \leq t < k)$ is the indication function that when $0 \leq t < k$, $I(t) = 1$ and when $t < 0$ or $k \geq k$, $I(t) = 0$. Continue analyzing the above inequality

$$
\begin{aligned}
\sum_{t=0}^\infty \left| (1-\gamma)\gamma^t - \frac{I(0 \leq t < k)}{k} \right| &= \sum_{t=0}^{k-1} \left| (1-\gamma)\gamma^t - \frac{1}{k} \right| + \sum_{t=k}^\infty (1-\gamma)\gamma^t \\
&= \sum_{t=0}^{k-1} \left| (1-\gamma)\gamma^t - \frac{1}{k} \right| + \gamma^k.
\end{aligned}
\tag{3}
$$

The above inequality (3) has different upper bound when 1) $1 \le k \le \frac{1}{1-\gamma}$ and 2) $k > \frac{1}{1-\gamma}$. We first analyze the upper bound for (3) when $1 \le k \le \frac{1}{1-\gamma}$, which indicating $\frac{1}{k} = 1 - \gamma \ge (1-\gamma)\gamma^t$ for all $t > 0$:

$$\sum_{t=0}^{k-1} \left| (1-\gamma)\gamma^t - \frac{1}{k} \right| = \sum_{t=0}^{k-1} \left( \frac{1}{k} - (1-\gamma)\gamma^t \right) = 1 - (1 - \gamma^k) = \gamma^k.$$

So when $1 \le k \le \frac{1}{1-\gamma}$, we have

$$\left| \sum_{s'} \left( d_s^{\pi_{\theta_{old}}}(s') - d_{k,s}^{\pi_{\theta_{old}}}(s') \right) \sum_a \pi_\theta(a|s') A_\gamma^{\pi_{\theta_{old}}}(s', a) \right| \le 2\gamma^k \cdot \frac{2|R_{\max}|}{1-\gamma} = \frac{4\gamma^k|R_{\max}|}{1-\gamma}. \quad (4)$$

Now we consider $k > \frac{1}{1-\gamma}$, which means $\frac{1}{k} < 1 - \gamma$,

$$\sum_{t=0}^{k-1} \left| (1-\gamma)\gamma^t - \frac{1}{k} \right| \le \sum_{t=0}^{k-1} \max\left( (1-\gamma)\gamma^t, \frac{1}{k} \right) \le \sum_{t=0}^{k-1}(1-\gamma) = k(1-\gamma).$$

So when $k > \frac{1}{1-\gamma}$, we have

$$\left| \sum_{s'} \left( d_s^{\pi_{\theta_{old}}}(s') - d_{k,s}^{\pi_{\theta_{old}}}(s') \right) \sum_a \pi_\theta(a|s') A_\gamma^{\pi_{\theta_{old}}}(s', a) \right| \le (k(1-\gamma) + \gamma^k) \cdot \frac{2|R_{\max}|}{1-\gamma}. \quad (5)$$

Now we start to analyze the second term in the RHS of equation (2)

$$\left| \sum_{s'} d_{k,s}^{\pi_{\theta_{old}}}(s') \sum_a \pi_\theta(a|s') \left( A_\gamma^{\pi_{\theta_{old}}}(s', a) - A_k^{\pi_{\theta_{old}}}(s', a) \right) \right| \le \max_{s,a} \left| A_\gamma^{\pi_{\theta_{old}}}(s, a) - A_k^{\pi_{\theta_{old}}}(s, a) \right|.$$

For the RHS of the above inequality, we have

$$\left| A_\gamma^{\pi_{\theta_{old}}}(s, a) - A_k^{\pi_{\theta_{old}}}(s, a) \right|$$

$$= \left| \left( Q_\gamma^{\pi_{\theta_{old}}}(s, a) - \mathbb{E}_{\pi_{\theta_{old}}}\left[ Q_\gamma^{\pi_{\theta_{old}}}(s, a) \right] \right) - \left( Q_k^{\pi_{\theta_{old}}}(s, a) - \mathbb{E}_{\pi_{\theta_{old}}}\left[ Q_k^{\pi_{\theta_{old}}}(s, a) \right] \right) \right|$$

$$= \left| \left( Q_\gamma^{\pi_{\theta_{old}}}(s, a) - Q_k^{\pi_{\theta_{old}}}(s, a) \right) - \mathbb{E}_{\pi_{\theta_{old}}}\left[ Q_\gamma^{\pi_{\theta_{old}}}(s, a) - Q_k^{\pi_{\theta_{old}}}(s, a) \right] \right|$$

$$\le 2\max_{s,a} \left| Q_\gamma^{\pi_{\theta_{old}}}(s, a) - Q_k^{\pi_{\theta_{old}}}(s, a) \right|.$$

We further have

$$\left| Q_\gamma^{\pi_{\theta_{old}}}(s, a) - Q_k^{\pi_{\theta_{old}}}(s, a) \right|$$

$$= \left| \mathbb{E}\left[ \sum_{t=0}^\infty \gamma^t r(s_t, a_t) \middle| s_0 = s, a_0 = a \right] - \mathbb{E}\left[ \sum_{t=0}^{k-1} r(s_t, a_t) \middle| s_0 = s, a_0 = a \right] \right|$$

$$= \left| \mathbb{E}\left[ \sum_{t=0}^{k-1} (\gamma^t - 1)r(s_t, a_t) + \sum_{t=k}^\infty \gamma^t r(s_t, a_t) \middle| s_0 = s, a_0 = a \right] \right|$$

$$\le \mathbb{E}\left[ \sum_{t=0}^{k-1} (1-\gamma^t)|r(s_t, a_t)| + \sum_{t=k}^\infty \gamma^t |r(s_t, a_t)| \middle| s_0 = s, a_0 = a \right]$$

$$\le \left( \sum_{t=0}^{k-1} (1-\gamma^t) + \sum_{t=k}^\infty \gamma^t \right) |R_{\max}|$$

$$\overset{(*)}{\le} \left( \sum_{t=0}^{k-1} t(1-\gamma) + \frac{\gamma^k}{1-\gamma} \right) |R_{\max}|$$

$$= \left( (1-\gamma)\frac{k(k-1)}{2} + \frac{\gamma^k}{1-\gamma} \right) |R_{\max}|.$$

The inequality $(*)$ holds because $t(1-\gamma) \geq (1-\gamma^t)$ for all $t \geq 0$. So we have

$$
\left| \sum_{s'} d_{k,s}^{\pi_{\theta_{\text{old}}}}(s') \sum_{a} \pi_\theta(a|s') \left( A_\gamma^{\pi_{\theta_{\text{old}}}}(s',a) - A_k^{\pi_{\theta_{\text{old}}}}(s',a) \right) \right|
$$

$$
\leq 2 \left( (1-\gamma)\frac{k(k-1)}{2} + \frac{\gamma^k}{1-\gamma} \right) |R_{\max}|.
$$

(6)

For $1 \leq k \leq \frac{1}{1-\gamma}$, combining (2), (4) and (6), we have

$$
|\mathcal{L}_\gamma(\pi_\theta, \pi_{\theta_{\text{old}}}, s) - \mathcal{L}_k(\pi_\theta, \pi_{\theta_{\text{old}}}, s)| \leq \frac{4\gamma^k |R_{\max}|}{1-\gamma} + 2 \left( (1-\gamma)\frac{k(k-1)}{2} + \frac{\gamma^k}{1-\gamma} \right) |R_{\max}|
$$

$$
= \frac{6\gamma^k |R_{\max}|}{1-\gamma} + |R_{\max}|(1-\gamma)k(k-1).
$$

To analyze the dominance term in the above inequality, we further denote $f(k) = \frac{6\gamma^k}{1-\gamma} + (1-\gamma)k(k-1)$. We can easily verify that when $\gamma^{1/(1-\gamma)} > 1/3$, $f(k+1) - f(k) = 2k(1-\gamma) - 6\gamma^k < 0$ for all $1 \leq k \leq \frac{1}{1-\gamma}$. So the term $\frac{6\gamma^k |R_{\max}|}{1-\gamma}$ is the dominate term, then we have

$$
|\mathcal{L}_\gamma(\pi_\theta, \pi_{\theta_{\text{old}}}, s) - \mathcal{L}_k(\pi_\theta, \pi_{\theta_{\text{old}}}, s)| \leq \mathcal{O}\left( \frac{\gamma^k}{1-\gamma} \right).
$$

For $k > \frac{1}{1-\gamma}$, combining (2), (5) and (6), we have

$$
|\mathcal{L}_\gamma(\pi_\theta, \pi_{\theta_{\text{old}}}, s) - \mathcal{L}_k(\pi_\theta, \pi_{\theta_{\text{old}}}, s)|
$$

$$
\leq (k(1-\gamma) + \gamma^k) \cdot \frac{2|R_{\max}|}{1-\gamma} + 2 \left( (1-\gamma)\frac{k(k-1)}{2} + \frac{\gamma^k}{1-\gamma} \right) |R_{\max}|
$$

$$
= 2k|R_{\max}| + |R_{\max}|(1-\gamma)k(k-1) + \frac{4\gamma^k |R_{\max}|}{1-\gamma}.
$$

To analyze the dominance term in the above inequality, we still define $f(k) = 2k + \frac{4\gamma^k}{1-\gamma} + (1-\gamma)k(k-1)$, it is easy to verify that $f(k+1) - f(k) = 2 + 2k(1-\gamma) - 4\gamma^k > 0$ for all $k > 1/(1-\gamma)$ and the first two terms in the RHS of the above inequality is the dominance term. So we have

$$
|\mathcal{L}_\gamma(\pi_\theta, \pi_{\theta_{\text{old}}}, s) - \mathcal{L}_k(\pi_\theta, \pi_{\theta_{\text{old}}}, s)| \leq \mathcal{O}(k + (1-\gamma)k(k-1)).
$$

Combine the above two cases, we finish the proof. $\qquad\square$

## D  PROOF OF LEMMA 2

**Lemma 2** (Performance difference under stationary state distribution). *Suppose Assumption 1 holds. For any two policies $\pi$, $\pi'$, the performance difference is given by*

$$
J_k(\pi) - J_k(\pi') = k \cdot \mathbb{E}_{s \sim d^\pi, a \sim \pi} \left[ A_k^{\pi'}(s,a) \right] + \sum_s \left( d^\pi(s) - d^{\pi'}(s) \right) V_k^{\pi'}(s) + \mathbb{E}_{s \sim d^\pi}[\Delta(s)],
$$

*where $\Delta(s) = \mathbb{E}_{\pi,P} \left[ \sum_{t=1}^{k-1} \mathbb{E}_{\pi',P}[r(s_{t+k-1}, a_{t+k-1})|s_t] \big| s_0 = s \right] - \mathbb{E}_{\pi,P} \left[ V_{k-1}^{\pi'}(s_k) \big| s_0 = s \right]$.*

*Proof.* From the definition of $J_k(\pi)$, we have

$$
J_k(\pi) - J_k(\pi') = \mathbb{E}_{s \sim d^\pi}[V_k^\pi(s)] - \mathbb{E}_{s \sim d^{\pi'}}\left[ V_k^{\pi'}(s) \right]
$$

$$
= \mathbb{E}_{s \sim d^\pi}[V_k^\pi(s)] - \mathbb{E}_{s \sim d^\pi}\left[ V_k^{\pi'}(s) \right] + \mathbb{E}_{s \sim d^\pi}\left[ V_k^{\pi'}(s) \right] - \mathbb{E}_{s \sim d^{\pi'}}\left[ V_k^{\pi'}(s) \right]
$$

$$
= \mathbb{E}_{s \sim d^\pi}\left[ V_k^\pi(s) - V_k^{\pi'}(s) \right] + \mathbb{E}_{s \sim d^\pi}\left[ V_k^{\pi'}(s) \right] - J_k(\pi')
$$

$$
\overset{(*)}{=} \mathbb{E}_{s \sim d^\pi}\left[ k \cdot \mathbb{E}_{s' \sim d_{k,s}^\pi, a \sim \pi}\left[ A_k^{\pi'}(s',a) \right] + \Delta \right] + \sum_s \left( d^\pi(s) - d^{\pi'}(s) \right) V_k^{\pi'}(s)
$$

$$
= k \mathbb{E}_{s \sim d^\pi, a \sim \pi}\left[ A_k^{\pi'}(s,a) \right] + \mathbb{E}_{s \sim d^\pi}[\Delta(s)] + \sum_s \left( d^\pi(s) - d^{\pi'}(s) \right) V_k^{\pi'}(s).
$$

The equality $(*)$ holds because of Lemma 1. This finish the proof. $\qquad\square$

# E   PROOF OF LEMMA 3

Our objective is to derive a bound on the TV distance between the stationary state distributions induced by any two policies, $\pi$ and $\pi'$. The proof hinges on two key components.

First, we establish that the transition operator under policy $\pi$, $P^\pi$, acts as a contraction with respect to the TV distance. This crucial contraction property is formalized in Lemma 8.

**Lemma 8** (Contraction property of total variation distance). *Suppose Assumption 1 and 2 holds. For two distribution $\mu$ and $\nu$ defined on the state space $S$, and a transition probability $P^\pi$ with policy $\pi$, we have*

$$D_{\text{TV}}(P^\pi\mu(\cdot), P^\pi\nu(\cdot)) \le \beta(\pi)D_{\text{TV}}(\mu(\cdot),\nu(\cdot)).$$

*Proof.* From the definition of $D_{\text{TV}}(P^\pi\mu(\cdot), P^\pi\nu(\cdot))$, we have

$$D_{\text{TV}}(P^\pi\mu(\cdot), P^\pi\nu(\cdot)) = \frac{1}{2}\sum_{s'}\left|\sum_s P^\pi(s'|s)\mu(s) - \sum_s P^\pi(s'|s)\nu(s)\right|$$

$$= \frac{1}{2}\sum_{s'}\left|\sum_s (\mu(s) - \nu(s))\,P^\pi(s'|s)\right|. \tag{7}$$

Now we define two sets $S^+ = \{s \in S | \mu(s) > \nu(s)\}$ and $S^- = \{s \in S | \mu(s) \le \nu(s)\}$, and because $\sum_s(\mu(s) - \nu(s)) = 0$, we have

$$\sum_{s\in S^+}(\mu(s) - \nu(s)) = -\sum_{s\in S^-}(\mu(s) - \nu(s)) = \sum_{s\in S^-}(\nu(s) - \mu(s)).$$

So we have $D_{\text{TV}}(\mu(\cdot),\nu(\cdot)) = \sum_{s\in S^+}(\mu(s) - \nu(s)) = \sum_{s\in S^-}(\nu(s) - \mu(s))$ and

$$\sum_s (\mu(s) - \nu(s))\,P^\pi(s'|s)$$

$$= \sum_{s_1\in S^+}(\mu(s_1) - \nu(s_1))\,P^\pi(s'|s_1) - \sum_{s_2\in S^-}(\nu(s_2) - \mu(s_2))\,P^\pi(s'|s_2)$$

$$= \sum_{s_1\in S^+}\sum_{s_2\in S^-}\frac{(\mu(s_1) - \nu(s_1))\,(\nu(s_2) - \mu(s_2))}{D_{\text{TV}}(\mu(\cdot),\nu(\cdot))}(P^\pi(s'|s_1) - P^\pi(s'|s_2)).$$

Combine the above relation with equation (7), and using triangle inequality, we can have

$$D_{\text{TV}}(P^\pi\mu(\cdot), P^\pi\nu(\cdot))$$

$$= \frac{1}{2}\sum_{s'}\left|\sum_{s_1\in S^+}\sum_{s_2\in S^-}\frac{(\mu(s_1) - \nu(s_1))\,(\nu(s_2) - \mu(s_2))}{D_{\text{TV}}(\mu(\cdot),\nu(\cdot))}(P^\pi(s'|s_1) - P^\pi(s'|s_2))\right|$$

$$\le \frac{1}{2}\sum_{s'}\sum_{s_1\in S^+}\sum_{s_2\in S^-}\frac{(\mu(s_1) - \nu(s_1))\,(\nu(s_2) - \mu(s_2))}{D_{\text{TV}}(\mu(\cdot),\nu(\cdot))}|(P^\pi(s'|s_1) - P^\pi(s'|s_2))|$$

$$= \sum_{s_1\in S^+}\sum_{s_2\in S^-}\frac{(\mu(s_1) - \nu(s_1))\,(\nu(s_2) - \mu(s_2))}{D_{\text{TV}}(\mu(\cdot),\nu(\cdot))}\left(\frac{1}{2}\sum_{s'}|(P^\pi(s'|s_1) - P^\pi(s'|s_2))|\right)$$

$$\le \sum_{s_1\in S^+}\sum_{s_2\in S^-}\frac{(\mu(s_1) - \nu(s_1))\,(\nu(s_2) - \mu(s_2))}{D_{\text{TV}}(\mu(\cdot),\nu(\cdot))}\left(\max_{s_1,s_2}\frac{1}{2}\sum_{s'}|(P^\pi(s'|s_1) - P^\pi(s'|s_2))|\right)$$

$$= \sum_{s_1\in S^+}\sum_{s_2\in S^-}\frac{(\mu(s_1) - \nu(s_1))\,(\nu(s_2) - \mu(s_2))}{D_{\text{TV}}(\mu(\cdot),\nu(\cdot))}\beta(\pi)$$

$$= \beta(\pi)D_{\text{TV}}(\mu(\cdot),\nu(\cdot)).$$

The proof is completed. $\square$

Second, we need to quantify how the transition dynamics differ under the two policies, which involves bounding the difference between their respective operators, $P^\pi$ and $P^{\pi'}$. This relationship is established in Lemma 9.

**Lemma 9** (Single step bias for two policies). *Suppose Assumption 1. For a given distribution $\mu$ defined on $S$ and two policies $\pi$, $\pi'$, we have*

$$D_{\text{TV}}(P^\pi \mu(\cdot), P^{\pi'} \mu(\cdot)) \leq \max_s D_{\text{TV}}(\pi(\cdot|s), \pi'(\cdot|s)).$$

*Proof.* From the definition of $D_{\text{TV}}(P^\pi \mu(\cdot), P^{\pi'} \mu(\cdot))$, we have

$$D_{\text{TV}}(P^\pi \mu(\cdot), P^{\pi'} \mu(\cdot)) = \frac{1}{2} \sum_{s'} \left| \sum_s P^\pi(s'|s)\mu(s) - \sum_s P^{\pi'}(s'|s)\mu(s) \right|$$

$$= \frac{1}{2} \sum_{s'} \left| \sum_s (P^\pi(s'|s) - P^{\pi'}(s'|s))\mu(s) \right|$$

$$\leq \sum_s \mu(s) \left( \frac{1}{2} \sum_{s'} \left| P^\pi(s'|s) - P^{\pi'}(s'|s) \right| \right)$$

$$\leq \sum_s \mu(s) \left( \max_s \frac{1}{2} \sum_{s'} \left| P^\pi(s'|s) - P^{\pi'}(s'|s) \right| \right)$$

$$= \max_s \frac{1}{2} \sum_{s'} \left| P^\pi(s'|s) - P^{\pi'}(s'|s) \right|.$$

Furthermore, we also have

$$\frac{1}{2} \sum_{s'} \left| P^\pi(s'|s) - P^{\pi'}(s'|s) \right| = \frac{1}{2} \sum_{s'} \left| \sum_a P(s'|s,a)\pi(a|s) - \sum_a P(s'|s,a)\pi'(a|s) \right|$$

$$= \frac{1}{2} \sum_{s'} \left| \sum_a P(s'|s,a)(\pi(a|s) - \pi'(a|s)) \right|$$

$$\leq \frac{1}{2} \sum_{s'} \sum_a P(s'|s,a) \left| \pi(a|s) - \pi'(a|s) \right|$$

$$\leq \frac{1}{2} \sum_a \sum_{s'} P(s'|s,a) \left| \pi(a|s) - \pi'(a|s) \right|$$

$$= \frac{1}{2} \sum_a |\pi(a|s) - \pi'(a|s)| = D_{\text{TV}}(\pi(\cdot|s), \pi'(\cdot|s)).$$

Combine the above two inequalities, we have the final result. $\qquad \square$

By combining the contraction property from Lemma 8 with the single-step policy difference from Lemma 9, we can then derive the final bound on the TV distance between two stationary state distributions with different policies.

**Lemma 3** (Distance between $d^\pi$ and $d^{\pi'}$). *Suppose Assumption 1 and 2 hold. For two policies $\pi$ and $\pi'$ with corresponding stationary distributions $d^\pi$ and $d^{\pi'}$, the total variation distance is bounded by:*

$$D_{\text{TV}}(d^\pi(\cdot), d^{\pi'}(\cdot)) \leq \frac{\max_s D_{\text{TV}}(\pi'(\cdot|s), \pi(\cdot|s))}{1 - \beta(\pi')}.$$

*Proof.* From the definition of stationary distribution, we have

$$D_{\text{TV}}(d^\pi(\cdot), d^{\pi'}(\cdot)) = D_{\text{TV}}(P^\pi d^\pi(\cdot), P^{\pi'} d^{\pi'}(\cdot))$$

$$= \frac{1}{2} \sum_{s'} \left| \sum_s P^\pi(s'|s)d^\pi(s) - \sum_s P^{\pi'}(s'|s)d^{\pi'}(s) \right|$$

$$= \frac{1}{2} \sum_{s'} \left| \sum_s P^\pi(s'|s) d^\pi(s) - \sum_s P^{\pi'}(s'|s) d^\pi(s) \right.$$

$$\left. + \sum_s P^{\pi'}(s'|s) d^\pi(s) - \sum_s P^{\pi'}(s'|s) d^{\pi'}(s) \right|$$

$$\leq \frac{1}{2} \sum_{s'} \left| \sum_s P^\pi(s'|s) d^\pi(s) - \sum_s P^{\pi'}(s'|s) d^\pi(s) \right|$$

$$+ \frac{1}{2} \sum_{s'} \left| \sum_s P^{\pi'}(s'|s) d^\pi(s) - \sum_s P^{\pi'}(s'|s) d^{\pi'}(s) \right|$$

$$= D_{\mathrm{TV}}(P^\pi d^\pi(\cdot), P^{\pi'} d^\pi(\cdot)) + D_{\mathrm{TV}}(P^{\pi'} d^\pi(\cdot), P^{\pi'} d^{\pi'}(\cdot))$$

$$\leq \max_s D_{\mathrm{TV}}(\pi(\cdot|s), \pi'(\cdot|s)) + \beta(\pi') D_{\mathrm{TV}}(d^\pi(\cdot), d^{\pi'}(\cdot)).$$

Rearrange the above inequality, we finally have

$$D_{\mathrm{TV}}(d^\pi(\cdot), d^{\pi'}(\cdot)) \leq \frac{\max_s D_{\mathrm{TV}}(\pi(\cdot|s), \pi'(\cdot|s))}{1 - \beta(\pi')} = \frac{\max_s D_{\mathrm{TV}}(\pi'(\cdot|s), \pi(\cdot|s))}{1 - \beta(\pi')}.$$

This complete the proof. □

## F  PROOF OF THEOREM 3

The proof of Theorem 3 largely follows the structure of the proof for its non-stationary counterpart, Theorem 1.

The central step is to establish an appropriate bound on the expected of the extra term, $\mathbb{E}_{s \sim d^\pi}[\Delta(s)]$. This bound is derived by combining the results of two preceding lemmas: Lemma 3, which bounds the distance between stationary distributions, and Lemma 8, which provides the contraction coefficient.

**Lemma 10.** *Suppose Assumption 1 and 2 hold. Given two policy $\pi$ and $\pi'$, and a initial state $s$, define $\Delta(s)$ as*

$$\Delta(s) = \mathbb{E}_{\pi, P} \left[ \sum_{t=1}^{k-1} \mathbb{E}_{\pi', P}[r(s_{t+k-1}, a_{t+k-1})|s_t] \middle| s_0 = s \right] - \mathbb{E}_{\pi, P} \left[ V_{k-1}^{\pi'}(s_k) \middle| s_0 = s \right],$$

*there are*

$$\mathbb{E}_{s \sim d^\pi}[\Delta(s)] \leq 2k \frac{\max_s D_{\mathrm{TV}}(\pi(\cdot|s), \pi'(\cdot|s))}{1 - \beta(\pi')} R_{\max}.$$

*Proof.* From the definition, we have

$$\mathbb{E}_{s \sim d^\pi} \left[ \mathbb{E}_{\pi, P} \left[ \sum_{t=1}^{k-1} \mathbb{E}_{\pi', P}[r(s_{t+k-1}, a_{t+k-1})|s_t] \right] \right]$$

$$= \sum_{t=0}^{k-2} \mathbb{E}_{s \sim d^\pi} \left[ \mathbb{E}_{\pi', P}[r(s_{k-1}, a_{k-1})|s_0 = s] \right]$$

$$= \sum_{t=0}^{k-2} \sum_{s_0} d^\pi(s_0) \sum_{s_{k-1}} P_{k-1}^{\pi'}(s_{k-1}|s_0) \sum_a \pi'(a|s_{k-1}) r(s_{k-1}, a),$$

and

$$\mathbb{E}_{s \sim d^\pi} \left[ \mathbb{E}_{\pi, P} \left[ V_{k-1}^{\pi'}(s_k) \middle| s_0 = s \right] \right] = \mathbb{E}_{s \sim d^\pi} \left[ V_{k-1}^{\pi'}(s) \right]$$

$$= \sum_{s_0} d^\pi(s_0) \sum_{t=0}^{k-2} \sum_{s_t} P_t^{\pi'}(s_t|s_0) \sum_a \pi'(a|s_t) r(s_t, a)$$

$$= \sum_{t=0}^{k-2} \sum_{s_0} d^{\pi}(s_0) \sum_{s_t} P_t^{\pi'}(s_t|s_0) \sum_a \pi'(a|s_t) r(s_t, a).$$

So we have

$$\mathbb{E}_{s \sim d^{\pi}}[\Delta(s)]$$

$$\leq \sum_{t=0}^{k-2} \left| \sum_{s_0} d^{\pi}(s_0) \sum_{s_{k-1}} P_{k-1}^{\pi'}(s_{k-1}|s_0) - \sum_{s_0} d^{\pi}(s_0) \sum_{s_t} P_t^{\pi'}(s_t|s_0) \right| R_{\max}$$

$$\leq \sum_{t=0}^{k-2} \sum_{s_0} \left| d^{\pi}(s_0) \sum_{s_{k-1}} P_{k-1}^{\pi'}(s_{k-1}|s_0) - d^{\pi}(s_0) \sum_{s_t} P_t^{\pi'}(s_t|s_0) \right| R_{\max}$$

$$= \sum_{t=0}^{k-2} \sum_{s_0} \left| d^{\pi}(s_0) \sum_{s_{k-1}} P_{k-1}^{\pi'}(s_{k-1}|s_0) - d^{\pi'}(s_0) + d^{\pi'}(s_0) - d^{\pi}(s_0) \sum_{s_t} P_t^{\pi'}(s_t|s_0) \right| R_{\max}$$

$$\leq \sum_{t=0}^{k-2} \left( \sum_{s_0} \left| d^{\pi}(s_0) \sum_{s_{k-1}} P_{k-1}^{\pi'}(s_{k-1}|s_0) - d^{\pi'}(s_0) \right| \right.$$

$$\left. + \sum_{s_0} \left| d^{\pi'}(s_0) - d^{\pi}(s_0) \sum_{s_t} P_t^{\pi'}(s_t|s_0) \right| \right) |R_{\max}|.$$

From Lemma 8, we have

$$\sum_{s_0} \left| d^{\pi}(s_0) \sum_{s_{k-1}} P_{k-1}^{\pi'}(s_{k-1}|s_0) - d^{\pi'}(s_0) \right| = (\beta(\pi'))^{k-1} D_{\text{TV}}(d^{\pi}(\cdot), d^{\pi'}(\cdot)),$$

and

$$\sum_{s_0} \left| d^{\pi'}(s_0) - d^{\pi}(s_0) \sum_{s_t} P_t^{\pi'}(s_t|s_0) \right| = (\beta(\pi'))^t D_{\text{TV}}(d^{\pi}(\cdot), d^{\pi'}(\cdot)).$$

Combine the above inequities, we have

$$\mathbb{E}_{s \sim d^{\pi}}[\Delta(s)] \leq |R_{\max}| \sum_{t=0}^{k-2} \left( (\beta(\pi'))^{k-1} + (\beta(\pi'))^t \right) D_{\text{TV}}(d^{\pi}(\cdot), d^{\pi'}(\cdot))$$

$$\overset{(*)}{\leq} 2k D_{\text{TV}}(d^{\pi}(\cdot), d^{\pi'}(\cdot)) |R_{\max}|$$

$$\overset{(**)}{\leq} 2k \frac{\max_s D_{\text{TV}}(\pi(\cdot|s), \pi'(\cdot|s))}{1 - \beta(\pi')} |R_{\max}|.$$

The inequality $(*)$ holds because of Assumption 2 and the inequality $(**)$ holds because of Lemma 3. This completes the proof. $\square$

With this bound on the expected difference established (Lemma 10 and Lemma 3), the remainder of the proof proceeds analogously to that of Theorem 1, completing the argument.

**Theorem 3** (Relation between the loss and performance difference under stationary state distribution). *Suppose Assumptions 1 and 2 hold. For a policy $\pi_\theta$ and a previous policy $\pi_{\theta_{old}}$, define the loss as:*

$$\mathcal{L}_k(\pi_\theta, \pi_{\theta_{old}}) = \mathbb{E}_{s \sim d^{\pi_{\theta_{old}}}, a \sim \pi_{\theta_{old}}} \left[ \frac{\pi_\theta(a|s)}{\pi_{\theta_{old}}(a|s)} A_k^{\pi_{\theta_{old}}}(s, a) \right],$$

*The relationship between policy improvement and this loss is bounded by:*

$$|J_k(\pi_\theta) - J_k(\pi_{\theta_{old}}) - k \cdot \mathcal{L}_k(\pi_\theta, \pi_{\theta_{old}})| \leq 4k \frac{D_{\text{TV}}^{\max}(\pi_{\theta_{old}} \| \pi_\theta)}{1 - \beta(\pi_{\theta_{old}})} \left( \alpha D_{\text{TV}}^{\max}(\pi_{\theta_{old}} \| \pi_\theta) + |R_{\max}| \right),$$

*where $D_{\text{TV}}^{\max}(\pi_{\theta_{old}} \| \pi_\theta) = \max_s D_{\text{TV}}(\pi_{\theta_{old}}(\cdot|s), \pi_\theta(\cdot|s))$ and $\alpha = \max_{s,a} \left| A_k^{\pi_{\theta_{old}}}(s, a) \right|$.*

*Proof.* Similar to the proof process in Theorem 1, we define

$$\xi(s) := \mathbb{E}_{a \sim \pi_{\theta_{\text{old}}}(\cdot|s)} \left[ \frac{\pi_\theta(a|s)}{\pi_{\theta_{\text{old}}}(a|s)} A_k^{\pi_{\theta_{\text{old}}}}(s,a) \right] = \mathbb{E}_{a \sim \pi_\theta(\cdot|s)} \left[ A_k^{\pi_{\theta_{\text{old}}}}(s,a) \right].$$

Because from the definition of $A^\pi(\cdot, \cdot)$, we have $\mathbb{E}_{a \sim \pi_{\theta_{\text{old}}}(\cdot|s)} \left[ A_k^{\pi_{\theta_{\text{old}}}}(s,a) \right] = 0$, for any $s$, we have

$$\begin{aligned}
|\xi(s)| &= \left| \sum_a \pi_\theta(a|s) A_k^{\pi_{\theta_{\text{old}}}}(s,a) - \sum_a \pi_{\theta_{\text{old}}}(a|s) A_k^{\pi_{\theta_{\text{old}}}}(s,a) \right| \\
&= \left| \sum_a \left( \pi_\theta(a|s) - \pi_{\theta_{\text{old}}}(a|s) \right) A_k^{\pi_{\theta_{\text{old}}}}(s,a) \right| \\
&\leq \sum_a |\pi_\theta(a|s) - \pi_{\theta_{\text{old}}}(a|s)| \max_{s,a} \left| A_k^{\pi_{\theta_{\text{old}}}}(s,a) \right| \\
&= 2\alpha D_{\text{TV}}^{\max}(\pi_{\theta_{\text{old}}} \| \pi_\theta),
\end{aligned}$$

where $D_{\text{TV}}^{\max}(\pi_{\theta_{\text{old}}} \| \pi_\theta) = \max_s D_{\text{TV}}(\pi_{\theta_{\text{old}}}(\cdot|s), \pi_\theta(\cdot|s))$ and $\alpha = \max_{s,a} \left| A_k^{\pi_{\theta_{\text{old}}}}(s,a) \right|$. Combine the above inequality with Lemma 3 and Lemma 10, we have

$$\begin{aligned}
&|J_k(\pi_\theta) - J_k(\pi_{\theta_{\text{old}}}) - k \cdot \mathcal{L}_k(\pi_\theta, \pi_{\theta_{\text{old}}})| \\
&= \left| k \sum_s \left( d^{\pi_\theta}(s) - d^{\pi_{\theta_{\text{old}}}}(s) \right) \xi(s) + \mathbb{E}_{s \sim d^{\pi_\theta}} \left[ V_k^{\pi_{\theta_{\text{old}}}}(s) \right] - J_k(\pi_{\theta_{\text{old}}}) + \mathbb{E}_{s \sim d^{\pi_\theta}} [\Delta(s)] \right| \\
&\leq \left| k \sum_s \left( d^{\pi_\theta}(s) - d^{\pi_{\theta_{\text{old}}}}(s) \right) \xi(s) + \sum_s \left( d^{\pi_\theta}(s) - d^{\pi_{\theta_{\text{old}}}}(s) \right) V_k^{\pi_{\theta_{\text{old}}}}(s) \right| + |\mathbb{E}_{s \sim d^{\pi_\theta}} [\Delta(s)]| \\
&= \left| \sum_s \left( d^{\pi_\theta}(s) - d^{\pi_{\theta_{\text{old}}}}(s) \right) \left( k\xi(s) + V_k^{\pi_{\theta_{\text{old}}}}(s) \right) \right| + |\mathbb{E}_{s \sim d^{\pi_\theta}} [\Delta(s)]| \\
&\leq \sum_s \left| \left( d^{\pi_\theta}(s) - d^{\pi_{\theta_{\text{old}}}}(s) \right) \left( k\xi(s) + V_k^{\pi_{\theta_{\text{old}}}}(s) \right) \right| + |\mathbb{E}_{s \sim d^{\pi_\theta}} [\Delta(s)]| \\
&\leq \sum_s |d^{\pi_\theta}(s) - d^{\pi_{\theta_{\text{old}}}}(s)| \left| k\xi(s) + V_k^{\pi_{\theta_{\text{old}}}}(s) \right| + |\mathbb{E}_{s \sim d^{\pi_\theta}} [\Delta(s)]| \\
&\leq \sum_s |d^{\pi_\theta}(s) - d^{\pi_{\theta_{\text{old}}}}(s)| \left( k|\xi(s)| + k|R_{\max}| \right) + |\mathbb{E}_{s \sim d^{\pi_\theta}} [\Delta(s)]| \\
&\leq 4k \frac{D_{\text{TV}}^{\max}(\pi_{\theta_{\text{old}}} \| \pi_\theta)}{1 - \beta(\pi_{\theta_{\text{old}}})} \left( \alpha D_{\text{TV}}^{\max}(\pi_{\theta_{\text{old}}} \| \pi_\theta) + |R_{\max}| \right).
\end{aligned}$$

This completes the proof. □

## G  PROOF OF THEOREM 4

The $J_k(\pi)$ has the following property.

**Lemma 11** (State value with $d^\pi$). *Suppose Assumption 1 holds, there is*

$$J_k(\pi) = \mathbb{E}_{s \sim d^\pi} [V_k^\pi(s)] = k \cdot \rho^\pi.$$

*Proof.* We first prove $\mathbb{E}_{s \sim d^\pi} [V_k^\pi(s)] = k \cdot \rho^\pi$. From the definition $\mathbb{E}_{s \sim d^\pi} [V_k^\pi(s)]$, we have

$$\mathbb{E}_{s \sim d^\pi} [V_k^\pi(s)] = \mathbb{E}_{s_0 \sim d^\pi, a_t \sim \pi(\cdot|s_t), s_{t+1} \sim P(\cdot|s_t, a_t)} \left[ \sum_{t=0}^{k-1} r(s_t, a_t) \middle| s_0 = s \right]$$

$$= \sum_s d^\pi(s) \sum_{t=0}^{k-1} \sum_{s'} P_t^\pi(s'|s) \sum_a \pi(a|s') r(s', a).$$

When $t = 0$, from the definition of $P_0(s'|s)$, we have

$$\sum_s d^\pi(s) \sum_{s'} P_0^\pi(s'|s) \sum_a \pi(a|s') r(s', a) = \sum_s d^\pi(s) \sum_a \pi(a|s) r(s, a) = \rho^\pi.$$

For any $1 \le t \le k - 1$, from the definition of stationary distribution, we further has the following relation

$$\sum_s d^\pi(s) P_t^\pi(s'|s) = \sum_s d^\pi(s) \sum_{s''} P_{t-1}(s'|s'') P(s''|s)$$

$$= \sum_{s''} \sum_s d^\pi(s) P(s''|s) P_{t-1}(s'|s'')$$

$$= \sum_{s''} d^\pi(s'') P_{t-1}(s'|s'')$$

$$= \sum_s d^\pi(s) P_{t-1}(s'|s).$$

The last equality holds by replacing the variable from $s''$ to $s$. Repeat the above process $t - 1$ times, we will then have

$$\sum_s d^\pi(s) P_t^\pi(s'|s) = \sum_s d^\pi(s) P^\pi(s'|s) = d^\pi(s').$$

So we have

$$\sum_s d^\pi(s) \sum_{s'} P_t^\pi(s'|s) \sum_a \pi(a|s') r(s', a) = \sum_{s'} d^\pi(s') \sum_a \pi(a|s') r(s', a) = \rho^\pi.$$

Sum all the above result, we finish the proof. $\qquad\square$

Then we represent the state value for $k$-sliding-window return as

$$V_k^\pi(s) := k \cdot \rho^\pi + \bar{V}^\pi(s) + \varepsilon_k^\pi(s), \text{ where } \varepsilon_k^\pi(s) = \mathbb{E}_{\pi, P} \left[ \sum_{t=k}^\infty (\rho^\pi - r(s_t, a_t)) \right]. \tag{8}$$

This bridge the gap between the state value function based on $k$-sliding-window return and average return. Besides, because $\mathbb{E}_{s \sim d^\pi}[V_k^\pi(s)] = k \cdot \rho^\pi$, we have $\mathbb{E}_{s \sim d^\pi}[\varepsilon_k^\pi(s)] = 0$. Using the above relationship, we can have Theorem 4.

**Theorem 4** (Difference between $\mathcal{L}_k(\pi_\theta, \pi_{\theta_{old}})$ and $\bar{\mathcal{L}}(\pi_\theta, \pi_{\theta_{old}})$)**.** *Suppose Assumptions 1 and 2 hold. Given $k$ and $0 < \xi_k \ll 1$ such that $k - 1 \ge T_{\mathrm{mix}}(\xi_k)$, the difference between the losses for policies $\pi_\theta$ and $\pi_{\theta_{old}}$ is bounded by:*

$$\left| \bar{\mathcal{L}}(\pi_\theta, \pi_{\theta_{old}}) - \mathcal{L}_k(\pi_\theta, \pi_{\theta_{old}}) \right| \le \mathcal{O} \left( \frac{D_{\mathrm{TV}}^{\max}(\pi_{\theta_{old}} \| \pi_\theta) \cdot \xi_k}{1 - \beta(\pi_{\theta_{old}})} \right),$$

*where $D_{\mathrm{TV}}^{\max}(\pi_{\theta_{old}} \| \pi_\theta) = \max_s D_{\mathrm{TV}}(\pi_{\theta_{old}}(\cdot|s), \pi_\theta(\cdot|s))$.*

*Proof.* From the definition of $\mathcal{L}_k(\pi_\theta; \pi_{\theta_{old}})$, and the new definition of $V_k^\pi(s)$ in equation 8, we have

$$\mathcal{L}_k(\pi_\theta, \pi_{\theta_{old}})$$

$$= \mathbb{E}_{s \sim d^{\pi_{\theta_{old}}}, a \sim \pi_{\theta_{old}}} \left[ \frac{\pi_\theta(a|s)}{\pi_{\theta_{old}}(a|s)} A_k^{\pi_{\theta_{old}}}(s, a) \right]$$

$$= \mathbb{E}_{s \sim d^{\pi_{\theta_{old}}}, a \sim \pi_{\theta_{old}}} \left[ \frac{\pi_\theta(a|s)}{\pi_{\theta_{old}}(a|s)} \left( Q_k^{\pi_{\theta_{old}}}(s, a) - V_k^{\pi_{\theta_{old}}}(s) \right) \right]$$

$$= \mathbb{E}_{s \sim d^{\pi_{\theta_{old}}}, a \sim \pi_{\theta_{old}}} \left[ \frac{\pi_\theta(a|s)}{\pi_{\theta_{old}}(a|s)} \left( r(s, a) + \mathbb{E}_{s' \sim P(\cdot|s,a)} \left[ V_{k-1}^{\pi_{\theta_{old}}}(s') \right] - V_k^{\pi_{\theta_{old}}}(s) \right) \right]$$

$$= \mathbb{E}_{s \sim d^{\pi_{\theta_{old}}}, a \sim \pi_{\theta_{old}}} \left[ \frac{\pi_\theta(a|s)}{\pi_{\theta_{old}}(a|s)} r(s, a) \right]$$

$$+ \mathbb{E}_{s \sim d^{\pi_{\theta_{old}}}, a \sim \pi_\theta, s' \sim P(\cdot|s,a)} \left[ V_{k-1}^{\pi_{\theta_{old}}}(s') \right] - \mathbb{E}_{s \sim d^{\pi_{\theta_{old}}}} \left[ V_k^{\pi_{\theta_{old}}}(s) \right]$$

$$
=\mathbb{E}_{s\sim d^{\pi_{\theta_{\mathrm{old}}}},a\sim\pi_{\theta_{\mathrm{old}}}}\left[\frac{\pi_\theta(a|s)}{\pi_{\theta_{\mathrm{old}}}(a|s)}r(s,a)\right]
$$

$$
+\mathbb{E}_{s\sim d^{\pi_{\theta_{\mathrm{old}}}},a\sim\pi_\theta,s'\sim P(\cdot|s,a)}\left[(k-1)\cdot\rho^{\pi_{\theta_{\mathrm{old}}}}+\bar{V}^{\pi_{\theta_{\mathrm{old}}}}(s')+\varepsilon_{k-1}^{\pi_{\theta_{\mathrm{old}}}}(s')\right]-k\cdot\rho^{\pi_{\theta_{\mathrm{old}}}}
$$

$$
=\mathbb{E}_{s\sim d^{\pi_{\theta_{\mathrm{old}}}},a\sim\pi_{\theta_{\mathrm{old}}}}\left[\frac{\pi_\theta(a|s)}{\pi_{\theta_{\mathrm{old}}}(a|s)}r(s,a)\right]
$$

$$
+\mathbb{E}_{s\sim d^{\pi_{\theta_{\mathrm{old}}}},a\sim\pi_\theta,s'\sim P(\cdot|s,a)}\left[\bar{V}^{\pi_{\theta_{\mathrm{old}}}}(s')+\varepsilon_{k-1}^{\pi_{\theta_{\mathrm{old}}}}(s')\right]-\rho^{\pi_{\theta_{\mathrm{old}}}}.
$$

Additionally, because $\mathbb{E}_{s\sim d^{\pi_{\theta_{\mathrm{old}}}}}[\bar{V}^{\pi_{\theta_{\mathrm{old}}}}(s)]=0$ and from the definition of the optimization objective of average reward, we have

$$
\bar{\mathcal{L}}(\pi_\theta,\pi_{\theta_{\mathrm{old}}})
$$

$$
=\mathbb{E}_{s\sim d^{\pi_{\theta_{\mathrm{old}}}},a\sim\pi_{\theta_{\mathrm{old}}}}\left[\frac{\pi_\theta(a|s)}{\pi_{\theta_{\mathrm{old}}}(a|s)}\bar{A}^{\pi_{\theta_{\mathrm{old}}}}(s,a)\right]
$$

$$
=\mathbb{E}_{s\sim d^{\pi_{\theta_{\mathrm{old}}}},a\sim\pi_{\theta_{\mathrm{old}}}}\left[\frac{\pi_\theta(a|s)}{\pi_{\theta_{\mathrm{old}}}(a|s)}\left(r(s,a)-\rho^{\pi_{\theta_{\mathrm{old}}}}+\mathbb{E}_{s'\sim P(\cdot|s,a)}[\bar{V}^{\pi_{\theta_{\mathrm{old}}}}(s')]-\bar{V}^{\pi_{\theta_{\mathrm{old}}}}(s)\right)\right]
$$

$$
=\mathbb{E}_{s\sim d^{\pi_{\theta_{\mathrm{old}}}},a\sim\pi_{\theta_{\mathrm{old}}}}\left[\frac{\pi_\theta(a|s)}{\pi_{\theta_{\mathrm{old}}}(a|s)}r(s,a)\right]
$$

$$
+\mathbb{E}_{s\sim d^{\pi_{\theta_{\mathrm{old}}}},a\sim\pi_\theta,s'\sim P(\cdot|s,a)}\left[\bar{V}^{\pi_{\theta_{\mathrm{old}}}}(s')\right]-\mathbb{E}_{s\sim d^{\pi_{\theta_{\mathrm{old}}}}}[\bar{V}^{\pi_{\theta_{\mathrm{old}}}}(s)]-\rho^{\pi_{\theta_{\mathrm{old}}}}
$$

$$
=\mathbb{E}_{s\sim d^{\pi_{\theta_{\mathrm{old}}}},a\sim\pi_{\theta_{\mathrm{old}}}}\left[\frac{\pi_\theta(a|s)}{\pi_{\theta_{\mathrm{old}}}(a|s)}r(s,a)\right]+\mathbb{E}_{s\sim d^{\pi_{\theta_{\mathrm{old}}}},a\sim\pi_\theta,s'\sim P(\cdot|s,a)}\left[\bar{V}^{\pi_{\theta_{\mathrm{old}}}}(s')\right]-\rho^{\pi_{\theta_{\mathrm{old}}}}.
$$

where the last inequality holds because of $\mathbb{E}_{s\sim d^{\pi_{\theta_{\mathrm{old}}}}}[\bar{V}^{\pi_{\theta_{\mathrm{old}}}}(s)]=0$. So we have

$$
\left|\bar{\mathcal{L}}(\pi_\theta,\pi_{\theta_{\mathrm{old}}})-\mathcal{L}_k(\pi_\theta,\pi_{\theta_{\mathrm{old}}})\right|=\left|\mathbb{E}_{s\sim d^{\pi_{\theta_{\mathrm{old}}}},a\sim\pi_\theta,s'\sim P(\cdot|s,a)}\left[\varepsilon_{k-1}^{\pi_{\theta_{\mathrm{old}}}}(s')\right]\right|.
$$

Because from the definition $\mathbb{E}_{s\sim d^\pi}[\varepsilon_{k-1}^\pi(s)]=0$, so we further have

$$
\left|\bar{\mathcal{L}}(\pi_\theta,\pi_{\theta_{\mathrm{old}}})-\mathcal{L}_k(\pi_\theta,\pi_{\theta_{\mathrm{old}}})\right|
$$

$$
=\left|\mathbb{E}_{s\sim d^{\pi_{\theta_{\mathrm{old}}}},a\sim\pi_\theta,s'\sim P(\cdot|s,a)}\left[\varepsilon_{k-1}^{\pi_{\theta_{\mathrm{old}}}}(s')\right]-\mathbb{E}_{s'\sim d^{\pi_{\theta_{\mathrm{old}}}}}[\varepsilon_{k-1}^{\pi_{\theta_{\mathrm{old}}}}(s')]\right|
$$

$$
=\left|\sum_{s'}\sum_s d^{\pi_{\theta_{\mathrm{old}}}}(s)\sum_a P(s'|s,a)\pi_\theta(a|s)\varepsilon_{k-1}^{\pi_{\theta_{\mathrm{old}}}}(s')-\sum_{s'}d^{\pi_{\theta_{\mathrm{old}}}}(s')\varepsilon_{k-1}^{\pi_{\theta_{\mathrm{old}}}}(s')\right| \tag{9}
$$

$$
\leq\left|\sum_{s'}d^{\pi_{\theta_{\mathrm{old}}}}(s')-\sum_{s'}\sum_s d^{\pi_{\theta_{\mathrm{old}}}}(s)\sum_a P(s'|s,a)\pi_\theta(a|s)\right|\max_{s'}\left|\varepsilon_{k-1}^{\pi_{\theta_{\mathrm{old}}}}(s')\right|.
$$

The difference between the distribution on the RHS can be bounded as

$$
\left|\sum_{s'}d^{\pi_{\theta_{\mathrm{old}}}}(s')-\sum_{s'}\sum_s d^{\pi_{\theta_{\mathrm{old}}}}(s)\sum_a P(s'|s,a)\pi_\theta(a|s)\right|
$$

$$
=\left|\sum_{s'}\sum_s d^{\pi_{\theta_{\mathrm{old}}}}(s)\sum_a P(s'|s,a)\pi_{\theta_{\mathrm{old}}}(a|s)-\sum_{s'}\sum_s d^{\pi_{\theta_{\mathrm{old}}}}(s)\sum_a P(s'|s,a)\pi_\theta(a|s)\right|
$$

$$
\leq\sum_{s'}\sum_s d^{\pi_{\theta_{\mathrm{old}}}}(s)\sum_a P(s'|s,a)\left|\pi_{\theta_{\mathrm{old}}}(a|s)-\pi_\theta(a|s)\right| \tag{10}
$$

$$
=\sum_s d^{\pi_{\theta_{\mathrm{old}}}}(s)\sum_a\sum_{s'}P(s'|s,a)\left|\pi_{\theta_{\mathrm{old}}}(a|s)-\pi_\theta(a|s)\right|
$$

$$
=\sum_s d^{\pi_{\theta_{\mathrm{old}}}}(s)\sum_a\left|\pi_{\theta_{\mathrm{old}}}(a|s)-\pi_\theta(a|s)\right|
$$

$$
=2\mathbb{E}_{s\sim d^\pi}\left[D_{\mathrm{TV}}(\pi_{\theta_{\mathrm{old}}}(\cdot|s),\pi_\theta(\cdot|s))\right]
$$

$$
\leq 2\max_s D_{\mathrm{TV}}(\pi_{\theta_{\mathrm{old}}}(\cdot|s),\pi_\theta(\cdot|s)).
$$

From the definition of $\varepsilon_{k-1}^{\pi_{\theta_{\text{old}}}}(s)$ and the fact that $\mathbb{E}_{s \sim d^{\pi_{\theta_{\text{old}}}}}[\varepsilon_{k-1}^{\pi_{\theta_{\text{old}}}}(s)] = 0$, we have

$$
\varepsilon_{k-1}^{\pi_{\theta_{\text{old}}}}(s) = \sum_{t=k-1}^{\infty} \sum_{s'} P_t^{\pi_{\theta_{\text{old}}}}(s'|s) \sum_a \pi_{\theta_{\text{old}}}(a|s')(\rho^{\pi_{\theta_{\text{old}}}} - r(s', a)) - \mathbb{E}_{s \sim d^{\pi_{\theta_{\text{old}}}}}[\varepsilon_{k-1}^{\pi_{\theta_{\text{old}}}}(s)]
$$

$$
= \sum_{t=k-1}^{\infty} \sum_{s'} P_t^{\pi_{\theta_{\text{old}}}}(s'|s) \sum_a \pi_{\theta_{\text{old}}}(a|s')(\rho^{\pi_{\theta_{\text{old}}}} - r(s', a))
$$

$$
- \sum_{t=k-1}^{\infty} \sum_{s'} d^{\pi_{\theta_{\text{old}}}}(s') \sum_a \pi_{\theta_{\text{old}}}(a|s')(\rho^{\pi_{\theta_{\text{old}}}} - r(s', a))
$$

$$
= \sum_{t=k-1}^{\infty} \sum_{s'} (P_t^{\pi_{\theta_{\text{old}}}}(s'|s) - d^{\pi_{\theta_{\text{old}}}}(s')) \sum_a \pi_{\theta_{\text{old}}}(a|s')(\rho^{\pi_{\theta_{\text{old}}}} - r(s', a)).
$$

From the definition of $P_t^{\pi_{\theta_{\text{old}}}}(s'|s)$ and $d^{\pi_{\theta_{\text{old}}}}(s')$, we have

$$
P_t^{\pi_{\theta_{\text{old}}}}(s'|s) - d^{\pi_{\theta_{\text{old}}}}(s') = \sum_{s''} P^{\pi_{\theta_{\text{old}}}}(s'|s'') P_{t-1}^{\pi_{\theta_{\text{old}}}}(s''|s) - \sum_{s''} P^{\pi_{\theta_{\text{old}}}}(s'|s'') d^{\pi_{\theta_{\text{old}}}}(s'').
$$

So from Lemma 8 and the assumption $k - 1 \geq T_{\text{mix}}(\xi_k)$, we have

$$
D_{\text{TV}}(P_t^{\pi_{\theta_{\text{old}}}}(\cdot|s), d^{\pi_{\theta_{\text{old}}}}(\cdot)) \leq \beta(\pi_{\theta_{\text{old}}}) D_{\text{TV}}(P_{t-1}^{\pi_{\theta_{\text{old}}}}(\cdot|s), d^{\pi_{\theta_{\text{old}}}}(\cdot)),
$$

$$
\max_s D_{\text{TV}}(P_{k-1}^{\pi_{\theta_{\text{old}}}}(\cdot|s), d^{\pi_{\theta_{\text{old}}}}(\cdot)) \leq \xi_k.
$$

Because $0 < \beta(\pi_{\theta_{\text{old}}}) < 1$, so we can have

$$
\sum_{t=k-1}^{\infty} \sum_{s'} (P_t^{\pi_{\theta_{\text{old}}}}(s'|s) - d^{\pi_{\theta_{\text{old}}}}(s')) \leq \sum_{t=k-1}^{\infty} 2 \max_s D_{\text{TV}}(P_t^{\pi_{\theta_{\text{old}}}}(\cdot|s), d^{\pi_{\theta_{\text{old}}}}(\cdot))
$$

$$
\leq 2 \sum_{t=k-1}^{\infty} \beta^{t-k+1}(\pi_{\theta_{\text{old}}}) \xi_k = \frac{2\xi_k}{1 - \beta(\pi_{\theta_{\text{old}}})}.
$$

So we finally have

$$
\varepsilon_{k-1}^{\pi_{\theta_{\text{old}}}}(s) \leq \frac{2\xi_k}{1 - \beta(\pi_{\theta_{\text{old}}})} \left| \max_{s', a} (\rho^{\pi_{\theta_{\text{old}}}} - r(s', a)) \right|. \tag{11}
$$

Combine (9), (10) and (11), we finally have

$$
\left| \bar{\mathcal{L}}(\pi_\theta, \pi_{\theta_{\text{old}}}) - \mathcal{L}_k(\pi_\theta, \pi_{\theta_{\text{old}}}) \right| \leq \frac{4\xi_k D_{\text{TV}}^{\max}(\pi_{\theta_{\text{old}}} \| \pi_\theta)}{1 - \alpha(\pi_{\theta_{\text{old}}})} \left| \max_{s', a} (\rho^{\pi_{\theta_{\text{old}}}} - r(s', a)) \right|,
$$

where $D_{\text{TV}}^{\max}(\pi_{\theta_{\text{old}}} \| \pi_\theta) = \max_s D_{\text{TV}}(\pi_{\theta_{\text{old}}}(\cdot|s), \pi_\theta(\cdot, s))$. This completes the proof. $\qquad \square$

## H  STATE VISITATION PATTERNS

Figrue 4 visualizes the state visitation patterns corresponding to trajectories sampled using a near-optimal policy in the test environments used in Section 5. The results reveal a distinct dichotomy: the patterns for *Swimmer, Walker2d, HalfCheetah*, and *Hopper* are periodic, while those for *Reacher* and *Pusher* are aperiodic. Consequently, we identify the first group of environments as stationary and the second as non-stationary.

## I  EXPERIMENTAL SETUP

The hyperparameter configuration for the experiments in Section 5 is summarized in Table 1. Our choices are grounded in established best practices and designed to ensure a fair comparison.

The common parameters, applied to all PPO variants, are detailed first. Both the policy and value functions are parameterized by fully-connected networks. Following standard practice in on-policy deep reinforcement learning, the policy network employs a Tanh activation function. In contrast,

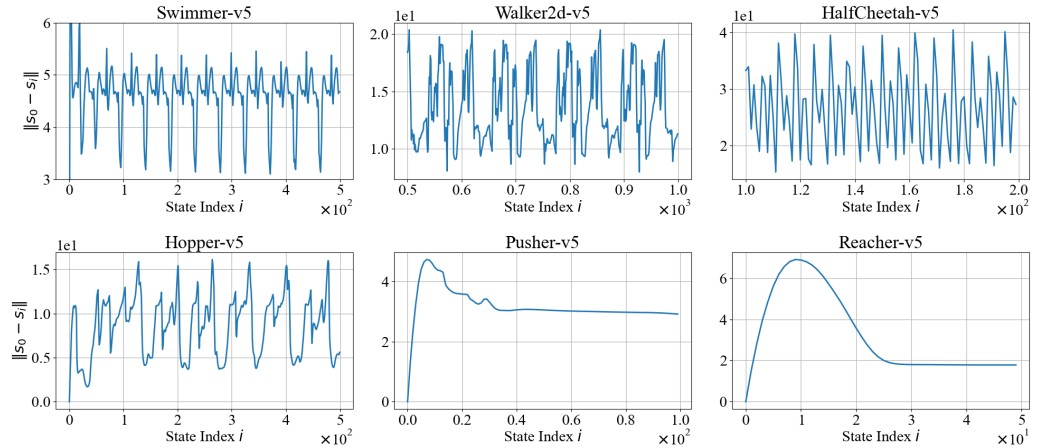

Figure 4: State visitation patterns for the environments in Section 5. Each plot shows the Euclidean distance from the initial state, $\|s_i - s_0\|$, as a function of the time step $i$ within a trajectory sampled from a near-optimal policy.

the value network uses ReLU, a choice made to enhance training stability and prevent vanishing gradients during value estimation.

For the average-return PPO baseline, we set the reset cost to zero. This decision is based on the findings of Zhang & Ross (2021), who demonstrated that the performance of average-return PPO/TRPO is largely insensitive to this specific parameter. This allows for a simpler implementation without sacrificing performance.

Table 1: Hyperparameter Setup

| Hyperparameter | Value |
|---|---|
| *Common Parameters* | |
| No. of training episodes | 2000 |
| Sample trajectories per episode | 5 |
| Max steps per trajectory | 1000 |
| Training epochs per episode | 10 |
| Mini-batch size | 256 |
| GAE parameter ($\lambda$) | 0.95 |
| PPO clipping ratio ($\epsilon$) | 0.2 |
| Entropy coefficient | 0.01 |
| Gradient clip norm | 0.5 |
| Hidden layers (policy & value) | 2 |
| Hidden units | 128 |
| Policy network activation | Tanh |
| Value network activation | ReLU |
| Optimizer | Adam |
| Learning rate (policy & value) | $2 \times 10^{-4}$ |
| *Average-Return PPO Specific* | |
| Gain $\hat{\rho}$ soft-update rate | 0.1 |
| Reset cost | 0 |

