# OpenReview forum: "A Unified Objective for On-Policy Reinforcement Learning in Stationary and Non-Stationary Environments"
_ICLR.cc/2026/Conference — ICLR 2026 Conference Withdrawn Submission_

### Official Review · Reviewer_Sq2C · 2025-10-30

**Soundness:** 1
**Presentation:** 3
**Contribution:** 1
**Rating:** 2
**Confidence:** 4

**Summary:**

The paper proposes a new objective, k-sliding window cumulative rewards, for learning the optimal policies in both “stationary” and “non-stationary” environments. The paper starts by pointing out the main difference between Swimmer and Reacher environments and then formally defines the k-sliding window objective for PPO algorithm. This objective is further compared with the discounted objective and the average reward objective.

**Strengths:**

The paper is well-written and easy to understand. The motivation example clearly illustrates the “stationary” and “nonstationary” environments and how the performance of policy differs if different objectives are used to optimize policies. The paper presents both theoretical and empirical analysis of the proposed objective.

**Weaknesses:**

I have some major concerns on the technical soundness and evaluation rigorousness of the paper in its current version.
### Lack of technical soundness
There might be some errors in the assumptions and theorems. Some theoretical analysis may not be sound:
1. In assumption 1, Ergodic means $P^\pi$ is **irreducible** and aperiodic.
2. **Theorem 1 may be wrong** (or there might be some constraints over k that hasn’t been specified correctly). One can verify this by setting $k=1$: the right hand side is now zero, which forces $V_k^{\pi}(s) – V_{k}^{\pi_{old}} = L_k(\pi, \pi_{old}, s)$. Based on the definition of $L_k$ in this theorem, $L_k$ differs from the first term on the right hand size of the equation in Lemma 1.
3. It’s unclear why the analysis focuses on the gaps of different losses? **The analysis in both Section 4.2 and 4.3 seems irrelevant if the new objective is introduced to optimize policies in both “stationary” and “nonstationary” environments**. Shouldn’t the analysis be focused on how the objective can be more effective than discounted return if the environments are “non-stationary”? Further, in Theorem 2, “the upper bound is minimized at $k=\frac{1}{1-\gamma}$, which may not imply anything about the policies learned by the new objective, as opposed to the discounted return objective. Likewise in Theorem 3, if “the magnitude of the policy update … diminishes” (which means $\pi_{old} = \pi$?) then both $\bar{L}$ and $L$ are effectively zero and the bound can be trivial. Also, why would it matter if “$L_k$ approximates $\bar{L}_k$? if the goal here is to approximate $\bar{L}_k$ then one can simply use the average return as the objective.


### Evaluation rigorousness
1. the main conclusion about the performance dichotomy between average and discounted return comes from **a very limited number of tests** on Swimmer and Reacher **only using PPO algorithm**. First, the reported performance on Swimmer-v5 in Figure 1(b) has a very large variance. It’s hard to draw the conclusion that “average-return PPO excels in the stationary Swimmer environment”. Even in Figure 2 (the main results), the variance of k-sliding-window and average remains very high. Second, since the paper only reports the results from PPO, it’s unclear if the performance gap reported in Figure 1(b) is really caused by the different objectives. It can be simply that hyperparameters in PPO are not robust to the change of objectives. The paper should reports more analysis with more different types of reinforcement learning methods.

2. in the experiment section, the proposed method benefits from some optimized tricks on GAE, L1 value loss, and Beta distribution, however **none of which have been ablated**. It would be a bit hard to see if the proposed objective can really yield a performant PPO and how these optimized tricks contribute to the final performance. Also **the baseline method can be quite weak**. The paper reported that “the vanilla PPO” has been used as the baseline for the discounted-return case. It seems this PPO baseline does not have any tricks used by the proposed method and can thus be in disadvantage. It’s also odd to not include a baseline simply because “Tang et al. (2021) lacks a public implementation”. The authors might need to reproduce Tang et al. (2021) for a comparison.

3. the theoretical analysis implies that when $k$ is set to $\frac{1}{1-\gamma}$, the concerned bounds “is minimized”. However, in the experiments, “k was set to 50 for Swimmer, Walker2d, and Hopper; 30 for HalfCheetah; and 10 for Reacher and Pusher.” **It seems the value of $k$ was somehow tuned for each environment, not consistent with the theoretical implications**. The ablation study on k also does not reflect this.

**Questions:**

1. Line 151: “da” is not needed
2. Line 210: $d^\pi_{k, s}(s’)$ may not be a probability distribution as it’s not normalized. (it used to be called state visitation measures)
3. Line 211 to 214: I guess this lemma can only be applied to $k\geq 2$? Otherwise the summation index would be problematic.
4. Line 353: “its unbiased nature …” why the objective is unbiased and in what sense?
5. Line 355: “structural similarities with the discounted return…” what are the structural similarities?
6. There are some terminology issues: I think it’s a bit odd to call Swimmer environment stationary and Reacher environment non-stationary. Clearly, both environments are stationary since the underlying transition dynamics and reward functions don’t change over time (namely the simulation itself is not changing over time). I guess the authors meant that the states in these two environments are of different types: the states in Swimmer are recurrent while the states in Reacher are transient?
7. “discount factor $\gamma$ inherently biases the objective towards short-term performance” but would k-sliding window then biases the objective towards k-steps performance?

---

> ### Author Response · Authors · 2025-11-25
>
> We are extremely grateful for your thorough and technically detailed review. Your sharp analysis has uncovered several critical errors and areas for improvement in our manuscript. We sincerely appreciate the time and effort you have invested, and we would like to address each of your points below.
>
> # Regarding Lack of Technical Soundness
>
> ## 1 & 2. About the Assumptions 1 and and the Theorem 1
> Thank you for your careful reading and for catching these errors. You are absolutely correct. The definition of Ergodic was a careless mistake on our part, and we will correct it. And the theorem implicitly requires the constraint $k ≥ 2$, and we sincerely apologize for failing to specify this crucial condition. We will make this constraint explicit in the revised manuscript.
>
> ## 3.  On the Rationale for Analyzing Loss Gaps (Sections 4.2 and 4.3)
> This is a crucial question regarding the motivation of our theoretical analysis. Our rationale is rooted in the fact that PPO is a **trust-region-type method**. The goal of the analysis in Section 4.3, culminating in Theorem 4, is to demonstrate that within such a policy optimization framework, the policy update derived from our **k-sliding-window objective** is almost equivalent to the update derived from a true **average-return objective**. This establishes a theoretical bridge, showing that our method is not just an arbitrary heuristic but can act as a practical proxy for average-reward optimization under specific conditions.
>
> However, we agree with your broader point. The current analysis does not sufficiently address why this objective is more effective in "non-stationary" environments, nor does it provide a formal analysis of the algorithm's convergence behavior. We acknowledge this is a significant gap, and we will add a more direct analysis from a convergence perspective in our future revisions to strengthen the paper's theoretical foundation.
>
> # Regarding  the Evaluation Rigorousness
>
> ## 1. Limited experiment on the motivation example
> We completely agree that our conclusions are drawn from a limited set of experiments and that the high variance you observed, especially in Figure 1(b) and Figure 2, weakens our claims. It is difficult to draw firm conclusions with such variance. We will expand our evaluation to include more on-policy algorithms and a significantly larger number of seeds to ensure our results are statistically robust.
>
> ## 2. On Unfair Baselines and Missing Ablation Studies
> Thank you for highlighting these critical flaws in our experimental setup. We will add a comprehensive ablation studys inthe future work.
>
> ## 3. On the Discrepancy Between Theoretical and Experimental $k$
> This is an excellent observation of a clear gap between our theory and our experimental practice. We will involve a deeper investigation into the relationship between the theoretically-implied optimal $k$ and the empirically found optimal $k$.

---

> ### Author Response · Authors · 2025-11-25
>
> # Questions
> 1. **Line 151**: Thank you, the typo "da" will be corrected.
> 2. **Line 210 ($d^{\pi}_{k,s}(s')$ as a distribution)**: Your concern is valid. However, in our definition, the sum of state visitations is divided by $k$. This division serves as a normalization factor, making $d^{\pi}_{k,s}(s')$ a valid probability distribution.
> 3. **Line 211-214 (Lemma scope)**: Yes, your understanding is correct. The lemma as written applies specifically to $k \geq 2$. We apologize for the ambiguity and will clarify the notation to make this clear.
> 4. **Line 353 ("unbiased nature")**: By "unbiased nature," we are referring to its relationship with the **undiscounted total return**, which is often the true metric for an agent's performance. The introduction of a discount factor $γ$ inherently biases the optimization objective away from this true evaluation metric. Our method, by not using $γ$, avoids this specific form of bias.
> 5. **Line 355 ("structural similarities")**: The "structural similarities" simply refer to the fundamental form of the objective as a **summation of a sequence of future rewards**. Both the k-sliding-window return and the discounted return are calculated via such a summation. We will rephrase this to be more precise.
> 6. **Terminology (Stationary vs. Non-stationary)**: We sincerely apologize for this major source of confusion. This was an incorrect use of terminology on our part. Our intention was to distinguish between environments where a given policy can induce a **stationary state distribution** (e.g., recurrent states in Swimmer) versus those where it cannot (e.g., transient states in Reacher). Our hypothesis is that the average-reward objective excels in the former case. We will correct this terminology throughout the entire manuscript.
> 7. **Bias of *k*-sliding-window**: This is a key question. While it may seem that a k-window introduces a $k$-step bias, our analysis in Theorem 4 suggests that when a stationary state distribution exists, our objective leads to policy updates that are effectively equivalent to those from an average-return objective. This indicates that it can approximate the behavior of a truly **long-horizon** objective, rather than simply being biased towards k-steps.
>
> Once again, we are profoundly grateful for your detailed and expert feedback. You have given us a clear and actionable path to significantly improve the technical soundness and empirical rigor of our work.

---

### Official Review · Reviewer_ASYF · 2025-11-01

**Soundness:** 2
**Presentation:** 3
**Contribution:** 3
**Rating:** 2
**Confidence:** 4

**Summary:**

The paper attempts to tackle a fundamental issue of RL: the dichotomy between discounted and average-return objectives. The proposed $k$-sliding-window return is an intuitive concept that defines the sum of the next $k$ rewards as the value function. The paper provides bounds on the loss difference between using $k$-step return and previous returns when updating the same policy. Additionally, the proposed algorithm, kSW-PPO, which incorporates the $k$-step return into PPO, exhibits comparable performance on several MuJoCo tasks.

**Strengths:**

The paper aims to address a very fundamental problem, providing an intuitive solution.

The algorithm and result are clearly explained.

All equations are clearly presented with well-defined symbols.

**Weaknesses:**

The fundamental definition of the task is not clear. What are stationary and non-stationary environments?

The newly introduced hyperparameter is not robust across tasks, requiring the same tuning efforts to choose between average return and discounted return.

The theoretical result does not provide an analysis of the algorithm's convergent behavior. (Check the Question section.)

**Questions:**

1. What are the definitions of stationary and non-stationary environments?

2. Does kSW-PPO converge? What are the bounds of the final returns compared to the average return or the discounted return? What convergent behaviour can we conclude from the loss bounds?

3. The paper reports the number of episodes. How many training steps does it correspond to?

---

> ### Author Response · Authors · 2025-11-25
>
> Thank you for your thoughtful review and for raising these important questions. Your feedback is extremely valuable and helps us identify the key areas where our manuscript needs improvement. We would like to address your specific concerns below.
>
> ## 1. Regarding the Definition of Stationary and Non-Stationary Environments:
> We sincerely apologize for the lack of clarity in our manuscript regarding these terms. You have correctly pointed out that our definition was not clear. This was a result of an imprecise use of terminology on our part.
>
> To clarify, our intention was not to distinguish between environments with fixed versus changing transition dynamics. Instead, we aimed to differentiate between scenarios based on whether a given policy could induce a **stationary state distribution**. Our underlying hypothesis is that the **average return** objective is more suitable for policies that can lead to such a stationary state distribution, whereas the **discounted return** objective is more effective in situations where a stationary state distribution cannot be established. We are grateful for your feedback, which highlights the need for us to correct this fundamental definition in the paper.
>
> ## 2. Regarding the Robustness of the Hyperparameter $k$
> We agree with your observation that the optimal value of $k$ varies across different environments. However, we would like to respectfully argue that tuning a single hyperparameter ($k$) within a unified framework is often a more straightforward and less structurally disruptive process than switching between two fundamentally different optimization objectives. Our method offers a way to interpolate between these two paradigms, and we see the tuning of $k$ as the mechanism for this adaptation.
>
> ## 3. Regarding the Convergence Analysis
> This is a critical point. In future work, we will dedicate our efforts to providing a rigorous analysis of the algorithm's convergence behavior and establishing bounds on its final return in comparison to the standard average return and discounted return methods.
>
> ## 4. Regarding the Relationship Between Episodes and Training Steps
> Thank you for asking for this clarification. In our experimental setup, the hyperparameters were configured such that for **every single trajectory sampled, we perform 10 parameter updates**. We will ensure this information is stated clearly in the experimental details section of the manuscript.
>
> Once again, we thank you for your time and constructive criticism. Your insights have provided us with a clear direction for improving the quality and rigor of our work.

---

### Official Review · Reviewer_JvrK · 2025-11-01

**Soundness:** 1
**Presentation:** 2
**Contribution:** 1
**Rating:** 2
**Confidence:** 4

**Summary:**

The paper notes that discounted objectives are better suited for non-stationary settings, while averag- reward objectives are better for stationary ones. They unify the two objectives via the "k-sliding window return", which aims to maximize the sum of rewards over a fixed horizon into the future. They provide analysis around how this objective compares with discounted and average-reward objectives. They then extend PPO to use this objective (kSW-PPO), and evaluate their extension against discounted and average-reward versions of PPO empirically.

**Strengths:**

* The contextualization in stationary and non-stationary settings is novel and one that isn't acknowledged often when comparing discounted vs. average-reward criteria.

* The analyses, as far as I checked, appear correct. The result in Section 4.3 where—in a stationary setting—the $k$-sliding-window leads to the same optimal policy as the average-reward criterion under the objective $\mathbb{E}_{s\sim d^\pi}[V_k^\pi(s)]$, was particularly interesting, and seems to mirror a comparable result for discounted objectives (Sutton & Barto, 2018; Naik et al., 2019).

### References

* Sutton, R. S., Barto, A. G. (2018). Reinforcement learning: an introduction.
* Naik, A., Shariff, R., Yasui, N., Yao, H., Sutton, R. S. (2019). Discounted reinforcement learning is not an optimization problem.

**Weaknesses:**

* The $k$-sliding-window return is not novel. In the RL space, De Asis et al. (2020) advocated for using an identical objective in non-finite-horizon problems, but motivated by how the objective leads to stable temporal difference updates. The idea goes farther back in the control theory literature—see receding horizon control.

* The analysis between the $k$-sliding-window and the discounted objective (Section 4.2) mirrors prior bias-variance analyses in the literature, as many such analyses would assume a finite-horizon to simplify things. The bias-equivalence between discounting and the sliding window emphasizes the commonly-acknowledged effective horizon interpretation of discounting, in terms of the expected sequence length of a stochastic process. This relationship highlights that $\gamma$ plays a very similar role as $k$, which kind of clashes with the claim that the $k$-sliding-window unifies discounted and average-reward objectives. That is, discounting already provides a way to interpolate toward the average-reward extreme, where crossing the Blackwell-optimal discount factor will lead to the same optimal policy as the average-reward criterion. Of note, $k$ interpolates horizontally while $\gamma$ interpolates vertically. This hard, horizontal cut-off prevents a guaranteed "Blackwell-optimal" $k$ where every $k$ beyond that would lead to the average-reward optimal policy. Instead, depending on the MDP, it can converge to a periodic loop where the optimal policy depends on $k$ modulo this period.

* Only 5 seeds were used in the empirical evaluation, which has been repeatedly shown to not be enough to make a proper statistical comparison for the claims being made (e.g., Henderson et al., 2017; Colas et al., 2018; Patterson et al, 2023; Patterson et al, 2024). Given the high variability of the results, it's unclear whether any conclusions drawn are within statistical significance. Of note, the shaded regions represent the standard deviation which is not a measure of confidence.

### References

* De Asis, K., Chan, A., Pitis, S., Sutton, R. S., Graves, D. (2020). Fixed-horizon temporal difference methods for stable reinforcement learning.
* Henderson, P., Islam, R., Bachman, P. Pineau, J., Precup, D., Meger, D. (2018). Deep reinforcement learning that matters.
* Colas, C., Sigaud, O., Oudeyer, P. (2018). How many random seeds? Statistical power analysis in deep reinforcement learning experiments.
* Patterson, A., Neumann, S., White, M., White, A. (2023). Empirical Design in Reinforcement Learning.
* Patterson, A., Neumann, S., Kumaraswamy, R., White, M., White, A. (2024). The Cross-environment Hyperparameter Setting Benchmark for Reinforcement Learning

**Questions:**

* The paper repeatedly mentions the "average return" objective, but I interpreted this to mean average reward given the context (e.g., the value definitions in 3.2 use differential returns, etc.). Can the authors clarify if this interpretation is correct? If so, I think it should definitely be revised as "average return" isn't well-situated in the literature.

* When mentioning how discounted objectives are better suited for non-stationary settings, what is this grounded in? On the theoretical side, the analyses of discounted algorithms often still rely on stationary assumptions for convergence. In a non-stationary problem, one could consider a hypothetical setup where an agent has access to the momentary optimal policies for both discounted and average-reward objectives—the average-reward optimal policy seems like it would lead to more total reward here, that the benefit of discounted objectives in such a setting may lie somewhere in between (perhaps on the algorithmic or bias-variance front).

* Regarding Section 4.3, can the authors comment on whether the same result can be achieved when comparing the same performance metric but with discounted values in place of the k-sliding window values? e.g., Section 10.4 of Sutton & Barto (2018) seems to arrive at this conclusion for the discounted version of the chosen performance metric.

* Can the authors comment on the statistical significance of the results, and whether 5 seeds are enough for the conclusions drawn?

---

> ### Author Response · Authors · 2025-11-25
>
> We would like to express our sincere gratitude for your meticulous review and for providing such deeply insightful and expert feedback. Your comments are invaluable and have prompted us to think more critically about the positioning and contributions of our work. We appreciate the opportunity to address the points you have raised.
>
> # Weakness:
>
> ## 1. On the Novelty of the k-Sliding-Window Objective
> Thank you for pointing out the connections to prior work, including receding horizon control and the work [1]. We acknowledge that the concept of a truncated or fixed-horizon objective is not entirely new.
>
> However, we would like to highlight a critical distinction in our approach compared to [1]. While their method also utilizes a truncated objective, it importantly retains a discount factor $\gamma < 1$. This inclusion of discounting, even within a fixed horizon, prevents the objective from truly approximating the undiscounted average-reward objective in the limit. As we argue in our paper, the presence of any discounting can introduce a bias that leads to suboptimal performance in environments where long-term reward accumulation is key (e.g., when a stationary state distribution can be induced by the policy). Our method's deliberate omission of this discount factor is a key design choice that allows Theorem 4 to hold, thus maintaining a principled connection to the average-reward objective, which we believe is a meaningful distinction.
>
> ## 2. On the Relationship with Blackwell Optimality and the Role of  $k$ vs. $γ$
> This is a very insightful and thought-provoking point, and we appreciate the depth of your analysis. We agree that adjusting the discount factor $\gamma$ towards 1 common approach for interpolating between average and discounted objectives, grounded in the strong theoretical framework of Blackwell optimality. And we are aware of the body of work that seeks to approximate the average-reward objective by using a discount factor $\gamma$ approaching 1 [2, 3], including the theoretical underpinnings of Blackwell optimality [4].
>
> However, frameworks based on Blackwell optimality often remain at a theoretical level without straightforward, practical algorithms. Other methods that bridge this gap can be considerably more complex to implement than simply modifying the optimization objective. Therefore, we position the contribution of our work as a **practical and accessible alternative**. We propose that tuning a single, intuitive hyperparameter $k$ (the window size) is arguably more straightforward for a practitioner than implementing a more complex algorithm or navigating the numerical and algorithmic subtleties of using near-unity discount factors.
>
>
> ## 3. On the Empirical Evaluation and Number of Seeds
> You have raised a very important point regarding the statistical rigor of our experiments. We will conduct a more thorough empirical evaluation with a significantly larger number of seeds to ensure that our conclusions are statistically significant.
>
> [1] De Asis, K., Chan, A., Pitis, S., Sutton, R. S., Graves, D. (2020). Fixed-horizon temporal difference methods for stable reinforcement learning.
> [2] Yunhao Tang, Mark Rowland, Remi Munos, and Michal Valko. Taylor expansion of discount factors.
> [3] Haichuan Gao, Zhile Yang, Tian Tan, Tianren Zhang, Jinsheng Ren, Pengfei Sun, Shangqi Guo, and Feng Chen. Partial consistency for stabilizing undiscounted reinforcement learning.
> [4] Manuel Schneckenreither. Average Reward Adjusted Discounted Reinforcement Learning Near-Blackwell-Optimal Policies for Real-World Applications

---

> ### Author Response · Authors · 2025-11-25
>
> # Questions
>
> ## 1. Clarification on "Average Return" vs. "Average Reward"
> Your interpretation is correct, and we sincerely apologize for this imprecise terminology. We will revise the manuscript thoroughly to use the correct and standard term "average reward" throughout to avoid any confusion.
>
> ## 2. Regarding the terminology of "Stationary" vs. "Non-Stationary" Environments
> We are grateful to you for pointing out the ambiguity in our use of these terms. We must clarify that this was a misapplication of terminology on our part. Our intention was not to refer to environments with changing transition probabilities (i.e., non-stationary in the conventional RL sense). Instead, we aimed to discuss whether a given policy could induce a stationary state distribution within the environment. We sincerely apologize for this confusion and appreciate your astuteness in identifying this critical issue. Your feedback highlights the need for greater precision in our manuscript.
>
> ## 3. Regarding the Performance Metric in Section 4.3
> Thank you for feedvack. The core of our analysis in Section 4.3 is predicated on using the **undiscounted total return** as the ultimate performance metric for evaluation. Our central argument is that when this is the metric of interest (as it often is in practical applications), any objective function that is inherently discounted will introduce a systematic bias. This bias is precisely what leads to suboptimal performance in settings where an undiscounted, average-reward approach is more appropriate (e.g., when a stationary state distribution exists).
>
> ## 4. On the Statistical Significance of the Results
> We will conduct a more thorough empirical evaluation with a significantly larger number of seeds to ensure that our conclusions are statistically significant.
>
> We are very grateful for your detailed and expert feedback. It has provided us with a clear roadmap for significantly improving the quality, rigor, and impact of our manuscript.

---

### Official Review · Reviewer_A5Ez · 2025-11-02

**Soundness:** 3
**Presentation:** 2
**Contribution:** 2
**Rating:** 2
**Confidence:** 3

**Summary:**

This paper proposes a novel objective for on-policy reinforcement learning, k-sliding-window return, which provides a trade-off between average- and discounted-reward returns. The authors further present a theoretical framework to validate and analyse its relationship with both average and discounted returns. Experimental results demonstrate that the proposed approach outperforms the selected baseline methods in both stationary and non-stationary environments.

**Strengths:**

The paper focuses on identifying a trade-off between average and discounted returns. To this end, a simple k-window approach is proposed and examined theoretically. Overall, the paper is mostly written and well organised. To the best of my knowledge, the proposed approach is novel.

**Weaknesses:**

I completely understand that the primary aim of this paper is to provide a theoretical framework. However, it currently lacks evidence demonstrating how the proposed approach could have a broader impact on the field.

First, the underlying motivation is only vaguely justified, relying on a single illustrative example (see Figure 1). Discounted approach is widely used in the literature. As discounted RL generally performs well in stationary environments, a deeper analysis would be valuable to clarify the necessity of a sliding-window approach. For instance, by discussing issues like bias.

Second, one would expect the proposed method to improve performance in non-stationary environments. However, the choice of baselines is rather limited, and the approach appears to perform better only in stationary settings, which calls the claimed trade-off into question (see Figure 2).

Third, the paper overlooks several relevant related works. Including comparisons with stronger and more representative baselines would considerably strengthen the paper’s contribution and positioning. For instance, discounting mismatch (https://www.ifaamas.org/Proceedings/aamas2022/pdfs/p1491.pdf), average reward adjusted discounted RL (https://arxiv.org/abs/2004.00857), and Trust Region Methods (https://arxiv.org/abs/2004.00857).

Fourth, my major concern is that the authors claim in the introduction that the environment cannot be known in advance. However, the optimal choice of the hyperparameter k appears to be environment-dependent (see Section 5.3). Specifically, k = 50 performs better in stationary environments, whereas k = 10 yields superior results in non-stationary settings. This observation seems to contradict the stated contributions of the proposed method.

**Questions:**

In addition to the points outlined above, I would like to raise a few further questions for clarification:

1) In Section 4.1, is the proposed method implemented using a sliding or an overlapping window? A sliding window may risk capturing only a limited time horizon. How do the authors ensure that the approach retains a sufficiently long-horizon perspective? Could an adaptive-window mechanism be employed to mitigate this limitation?

2) While PPO is one of the most widely adopted algorithms in the literature, could the proposed method be extended to other approaches, such as actor–critic frameworks? Is the method inherently specific to PPO, or could it be incorporated into alternative algorithms in a more plug-and-play fashion?

Overall, I would like to acknowledge the authors’ considerable effort in developing and presenting a solid theoretical framework. However, Sections 4.2 and 4.3 could be moved to the appendix. Instead, the authors could provide more substantial empirical evidence to demonstrate the broader and more universal benefits of the proposed approach.

---

> ### Author Response · Authors · 2025-11-25
>
> We sincerely thank you for your detailed feedback and insightful questions. Your comments are invaluable and have helped us to critically reflect on our work. We would like to take this opportunity to address the points you raised and provide further clarification.
>
> ## 1. Regarding the terminology of "Stationary" vs. "Non-Stationary" Environments
> We are grateful to you for pointing out the ambiguity in our use of these terms. We must clarify that this was a misapplication of terminology on our part. Our intention was not to refer to environments with changing transition probabilities (i.e., non-stationary in the conventional RL sense). Instead, we aimed to discuss whether a given policy could induce a **stationary state distribution** within the environment. We sincerely apologize for this confusion and appreciate your astuteness in identifying this critical issue. Your feedback highlights the need for greater precision in our manuscript.
>
> ## 2. Regarding the choice of hyperparameter $k$ and prior knowledge of the environment
> We concur with your observation that the optimal choice of the hyperparameter $k$ is indeed environment-dependent. However, we would like to respectfully argue that tuning a single, intuitive hyperparameter like $k$ is a considerably more straightforward and less costly procedure from a practical standpoint than changing an entire optimization objective. Our framework offers this flexibility through a tunable parameter, which we see as a practical advantage over methods that may require more fundamental architectural changes.
>
> ## 3. Clarification on the Windowing Mechanism (Section 4.1)
> Regarding the question about the **overlapping window**. From my prespective, these two terms are pretty similiar. To be more specific, for any given sampled trajectory, the empirical value functions estimated at consecutive timesteps, such as $V(s_t)$ and $V(s_{t+1})$, will share $k-1$ states. This overlap or sliding ensures a degree of continuity. Thank you for prompting this important clarification.
>
> ## 4. Regarding the Applicability of the Method Beyond PPO
> This is an excellent point. Because PPO itself is an actor-critic algorithm, we surmise that the question may be pointing towards its applicability to **off-policy** algorithms, as opposed to on-policy ones like PPO. Our proposed optimization objective is primarily designed for **on-policy** methods. Therefore, it can be integrated with other on-policy algorithms such as REINFORCE and A2C, but it is more suitable for trust-region-type on-policy algorithm, like TRPO and PPO.
>
> We sincerely appreciate the time and expertise you have dedicated to our work. Your efforts have been immensely helpful and have provided us with a much clearer path forward for this research.

---

### Note · Authors · 2025-12-01

**Comment:**

I would like to express my sincere gratitude to the program committee and the reviewers who dedicated their time and expertise to reviewing my manuscript. Their feedback is highly valued.

Upon reflection, I have concluded that the paper in its current state does not meet the standards I have set for my work and would benefit from significant revisions. Therefore, I have decided to withdraw it at this time to allow for further development.

Thank you for your understanding and for the opportunity to submit my work to your prestigious conference. I apologize for any inconvenience this may cause.

**Withdrawal Confirmation:**

I have read and agree with the venue's withdrawal policy on behalf of myself and my co-authors.